# Investigation of aerosol-cloud-rainfall association over Indian Summer Monsoon region

Chandan Sarangi[1], SachchidaNand Tripathi[1,*], Vijay. P. Kanawade[1,4], Ilan Koren[2], and

D. Sivanand Pai[3]

Affiliations:

[1]Department of Civil Engineering and Centre for Environmental Science and Engineering, India Institute of Technology, Kanpur, India

[2]Department of Earth and Planetary Science, Weizmann Institute, Rehovot, Israel

[3]India Meteorological Department, Pune, India

[4]Currently at University Centre for Earth and Space Sciences, University of Hyderabad, Hyderabad, India

*Correspondence to **snt@iitk.ac.in**

## **Abstract**

Monsoonal rainfall is the primary source of surface water in India. Using 12 years of in-situ and satellite observations, we examined association of aerosol loading with cloud fraction, cloud top pressure, cloud top temperature, and daily surface rainfall over Indian summer monsoon region (ISMR). Our results showed positive correlations between aerosol loading and cloud properties as well as rainfall. A decrease in outgoing longwave radiation and increase in reflected shortwave radiation at the top of the atmosphere with an increase in aerosol loading further indicates a possible seminal role of aerosols in deepening of cloud systems. Significant perturbation in liquid- and ice-phase microphysics was also evident over ISMR. For the polluted cases, delay in the onset of collision-coalescence processes and enhancement in the condensation efficiency, allows for more condensate mass to be lifted up to the mixed-colder phases. This results in the higher mass concentration of bigger sized ice-phase hydrometeors and, therefore, implies that the delayed rain processes eventually lead to more surface rainfall. Numerical simulation of a typical rainfall event case over ISMR using spectral bin microphysical scheme coupled with Weather Research Forecasting (WRF-SBM) model was also performed. Simulated microphysics also illustrated that the initial suppression of warm rain coupled with increase in updraft velocity under high aerosol loading leads to enhanced super-cooled liquid droplets above freezing level and ice-phase hydrometeors, resulting in increased accumulated surface rainfall. Thus, both observational and numerical analysis suggest that high aerosol loading may induce cloud invigoration and thereby increasing surface rainfall over the ISMR. While the meteorological variability influences the strength of the observed positive association, our results suggest that the persistent aerosol-associated deepening of cloud systems and intensification of surface rain amounts was applicable to all the meteorological sub-regimes over the ISMR. Hence, we believe that these results provide a step forward in our ability to address aerosol-cloud-rainfall associations based on satellite observations over ISMR

**Keywords:** Aerosol, cloud invigoration, rainfall, ISMR, WRF-SBM

**Introduction**

Aerosol-cloud-rainfall interactions and their feedbacks pose one of the largest uncertainties in understanding and estimating anthropogenic contribution of aerosols to climate forcing [*Forster et al.*, 2007; *Lohmann and Feichter*, 2005]. A fraction of aerosol particles gets activated as cloud condensation nuclei (CCN) to form the fundamental requisite for cloud droplet formation. Thus, perturbations in regional aerosol loading not only influence the radiation balance directly but also indirectly via perturbing the cloud properties and thereby the hydrological cycle[*Ramanathan et al.*, 2001].

Increase in aerosol loading near cloud base decreases the cloud droplet size and increases the cloud droplet number concentration[*Fitzgerald and Spyers-Duran*, 1973; *Squires*, 1958; *Squires and Twomey*, 2013; *Twomey*, 1974; 1977; *Warner and Twomey*, 1967] These microphysical changes initiate many feedbacks. The narrowing of the droplet size distribution was suggested to delay the onset of droplet collision-coalescence processes and thereby enhancing the cloud lifetime[*Albrecht*, 1989] and the delay of raindrop formation[*Khain*, 2009; *Rosenfeld*, 1999; 2000]. However, recent studies show that aerosol-induced initial stage suppression of raindrop formation provides the feedback mechanism for a change in microphysical-dynamical coupling within convective clouds, and results in the formation of deeper and wider invigorating clouds[*Andreae et al.*, 2004; *Koren et al.*, 2005]. For convective clouds with warm base, the activation and water supply all start in the warm part near the cloud base. The enhancement in droplet condensation releases more latent heat and, therefore, enhances updraft [*Dagan et al.*, 2015; *Pinsky et al.*, 2013; *Seiki and Nakajima*, 2014]. At the same time, smaller droplets will have smaller effective terminal velocity (i.e. better mobility) and, therefore, will be lifted higher in the atmosphere by the enhanced updrafts[*Heiblum et al.*, 2016; *Ilan et al.*, 2015]. Stronger updrafts and smaller effective terminal velocity result in more liquid mass being pushed up to the mixed and cold phases.

Smaller sized droplets will freeze higher in the atmosphere [*Rosenfeld and Woodley*, 2000] releasing the freezing latent heat in relatively colder environment, boosting the updrafts and further invigorating the cloud system [*O. Altaratz et al.*, 2014; *Andreae et al.*, 2004; *Khain et al.*, 2008; *Koren et al.*, 2005]. Hence, aerosol abundance can eventually cause intensification

of precipitation rate due to cloud invigorating effect under convective conditions[*Koren et al.*, 2014; *Koren et al.*, 2012; *Li et al.*, 2011].In contrast, under low cloud fraction condition, the presence of high concentration of absorbing aerosols induces aerosol semi-direct effect causing cloud inhibition[*Ackerman et al.*, 2000; *Koren et al.*, 2004; *Rosenfeld*, 1999] and thereby reduction in surface rainfall. Thus, the aerosol-cloud associations observed over any

given region is the net outcome of these competing aerosol effects on clouds[*Koren et al.*, 2008; *Rosenfeld et al.*, 2008]. Our present understanding of the sign as well as the magnitude of change in accumulated surface rainfall due to aerosols is inadequate. Besides, aerosol-cloud-rainfall associations are highly sensitive to variation in thermodynamical and environmental conditions, cloud properties, and aerosol types [*Khain et al.*, 2008; *Lee*, 2011;

*Tao et al.*, 2012], further complicating these interactions. Moreover, clouds and precipitation can also interact with aerosols through wet scavenging process [*Grandey et al.*, 2013; *Grandey et al.*, 2014; *Yang et al.*, 2016]. Global model simulations illustrated thatwet scavenging can cause a strong negative cloud fraction-AOD correlation over the tropics [*Grandey et al.*, 2013]. Wet scavenging effect can also generate similar negative rain rate-

AOD association in the tropical and mid-latitude oceans [*Grandey et al.*, 2014].

Indian summer monsoon is the lifeline for regional ecosystems and water resources, and plays a crucial role in India's agriculture and economy[*Webster et al.*, 1998]. Indian summer monsoon from June through September (JJAS) fulfils about 75% of the annual rainfall over central-north India. Variation in daily rainfall during summer monsoon rainfall is directly

linked to India's Kharif food grain production[*Preethi and Revadekar*, 2013]. A rapid

increase in population and industrialization over the last two decades has also resulted in high anthropogenic aerosol loading over Northern India, particularly in the Gangetic basin[*Dey and Di Girolamo*, 2011]. Consequently, the net impact of such large continental aerosol loading on cloud properties and daily surface rainfall in India is an important question that

requires utmost attention. Recent studies based on aerosol direct effect have shown different plausible pathways of aerosol impact on rainfall. Lau and Kim (2006)[*Lau and Kim*, 2006] have shown that aerosol-induced atmospheric heating over Himalayan slopes and Tibetan plateau during monsoon onset period, intensifies the northward shift of Indian summer monsoon, causing reduction in rainfall over ISMR. On the other hand, high aerosol loading

also induces solar dimming (absorbing) effect at surface [*Ramanathan and Carmichael*, 2008; *Ramanathan et al.*, 2001], which can alter the land-ocean thermal gradient and weaken the meridional circulation, resulting in drying trend in seasonal rainfall during Indian summer monsoon [*Bollasina et al.*, 2011; *Ganguly et al.*, 2012].Presence of higher concentrations of absorbing aerosols over North India is shown to induce a stronger north–south temperature

difference which fosters enhancement in moisture convergence from ocean and transition of a break spell of ISM into an active spell of ISM [*Manoj et al.*, 2011]. Further, this aerosol radiative effect causes increase in the moist static energy, invigoration of convection and eventually more rainfall over India during the following active phase[*Hazra et al.*, 2013; *Manoj et al.*, 2011]. These studies provide valuable insight on different pathways of aerosol's

radiative impact on the monsoon dynamics and seasonal rainfall over India. However, the microphysical aspect of aerosol's impact on the sign and the magnitude of the monsoonal rainfall over the Indian summer monsoon region (ISMR)is largely unknown [*Rosenfeld et al.*, 2014]. Nevertheless, a few recent studies have indicated existence of strong aerosol microphysical effect on cloud systems over ISMR [*Konwar et al.*, 2012; *Manoj et al.*, 2012;

*Prabha et al.*, 2012; *Sarangi et al.*, 2015; *Sengupta et al.*, 2013].Conversely, summer

monsoon plays an important role in determining variation in aerosol loading over India by brining clean marine air and wet scavenging, which are as important as emission in determining aerosol concentration [*Li et al.*, 2016]. It has also been shown that aerosols over the Indian Ocean interplay with seasonal changes over ISMR[*Corrigan et al.*, 2006].

Here, we have used 12 years (JJAS) of gridded datasets of surface rainfall, aerosol and cloud properties to examine aerosol-related changes in cloud macro-, micro- and radiative properties, and thereby on daily surface rainfall over ISMR. Aerosol associated changes in onset of warm rain, microphysical profiles and cloud radiative forcing isanalysed using observation and idealized simulations to investigate significance of aerosol microphysical effect over ISMR. The role of meteorology and aerosol humidification effect due to cloud contamination in retrieved aerosol optical depth (AOD)is also estimated to ensure the causality of the observed associations. This comprehensive effort to understand aerosol-cloud-rainfall interactions over India will likely illustrate the significance of aerosol's impact on monsoonal rainfall via microphysical pathway under continental conditions.

## 2. Data and methodology

### 2.1. Aerosol, cloud, rainfall, and radiation datasets

**Table 1**. Summary of daily dataset used in our analysis. LT refers to local time.

| Data source | Parameters | Temporal resolution (LT) | Time Period |
|---|---|---|---|
| IMD[*] | Accumulated rainfall | 08:30 am – 08:30 am | 2002-2013 |
| MODIS Aqua L3 (c5.1) | AOD, CF, CTT and CTP | 1:30 pm | 2002-2013 |
| CLOUDSAT[*] 2B (V8) | Mass concentration and effective radius of liquid- and ice-phase microphysical profiles | 1:30 pm | 2007-2011 |
| TRMM[*] 3B42 (V7) | Precipitation rate | 12:00 pm | 2002-2013 |
| NOAA-NCEP GDAS | Meteorological fields | 11:30 am | 2002-2013 |
| CERES L3 (Edition 3A) | TOA fluxes: SW (0-5 µm) and LW (5-100 µm) | 11:30 am – 2:30 pm | 2002-2013 |

| WMO Station Radiosondes | Temperature, Relative humidity, dew point | 5:30 am and 5:30 pm | 2002-2013 |
|---|---|---|---|

*Retrieved 0.25 deg. dataset re-gridded linearly to 1.0 deg. spatial resolution.

Table 1 summarizes in-situ and satellite observations used in this study . For correlation analysis between aerosol-cloud macrophysics, we used retrievals of AOD, cloud

fraction (CF), cloud top pressure (CTP) and cloud top temperature (CTT) from Moderate Resolution Imaging Spectro-radiometer (MODIS) onboard Aqua spacecraft [*Platnick et al.*, 2003; *Remer et al.*, 2005]. MODIS AOD has been validated extensively over land [*Remer et al.*, 2005; *Tripathi et al.*, 2005].

A new high resolution ($0.25^o \times 0.25^o$ gridded) daily rainfall (RF) dataset prepared by

India Meteorological Department (IMD) [*Pai et al.*, 2013] was used to represent accumulated surface rainfall. Quality assured measurements of RF from in-situ rain gauge stations (~6955) across the country were interpolated using an inverse distance weighted interpolation scheme [*Shepard*, 1968], to create this gridded product. The daily surface rainfall from previous day (08:30 am, local time) till 08:30 am (local time) present day has been recorded as daily

rainfall at all rain gauge stations maintained by IMD for 110 years (1901-2013). This product has been extensively validated against previous IMD rainfall products as well as the Asian Precipitation - Highly-Resolved Observational Data Integration Towards Evaluation (APHRODITE) rainfall dataset [*Pai et al.*, 2013]. IMD daily rainfall gridded datasets have been widely used by several investigators to study the rainfall climatology and its inter-

seasonal and intra-seasonal variability over Indian summer monsoon region [*Goswami et al.*, 2006; *Krishnamurthy and Shukla*, 2007; 2008; *Pai et al.*, 2014; *Rajeevan et al.*, 2008]. The precipitation rate (PR) at 12 PM local time was also obtained from the Tropical Rainfall Measuring Mission (TRMM) [*Huffman et al.*, 2010]. RF as well as PR datasets were linearly re-gridded to the $1^o \times 1^o$ grid for consistency in our correlation analysis.

For the correlation analysis between any two variables, only those spatio-temporal grids were considered where collocated measurements of both variables were available.The collocated variables RF, PR, CF, CTP and CTT were then sorted as a function of AOD and averaged to create total 50 scatter points. AODs > 1.0 (~ 5%)were omitted to reduce possibility of inclusion of cloud contaminated data in our analysis. Shallow clouds with CTP > 850 hPa (about 7 %) were also not considered in this analysis. Previous studies have also reported aerosol microphysical effect using such correlation analysis based on satellite datasets[*Chakraborty et al.*, 2016; *Feingold et al.*, 2001; *Kaufman et al.*, 2002; *Koren et al.*, 2010a; *Koren et al.*, 2014; *Koren et al.*, 2004; *Koren et al.*, 2012; *Myhre et al.*, 2007].Importantly, the availability of the ground based in-situ daily rainfall dataset enables us to further investigate the aerosol-cloud-rainfall association over ISMR spanning from 17° N to 27° N in latitude and 75° E to 88° E in longitude (bounded by black box in Figure 1). Here, we have excluded regions with mountainous terrain (Himalayan terrains to the north) and desert/barren land use regions (Thar Desert and nearby arid regions). This was done to avoid inclusion of extreme orographic precipitation as well as retrieval error in the satellite products (e.g. lower sensitivity over brighter land surfaces for MODIS aerosol products). ISMR has previously been extensively studied by several investigators [*Bollasina et al.*, 2011; *Goswami et al.*, 2006; *Sengupta et al.*, 2013] as the rainfall variability over this region is highly correlated with that of the entire India rainfall during June to September[*Gadgil*, 2003].Generally, aerosol loading over ISMR is very high (climatologically mean AOD of 0.56, Figure 1A), particularly over densely populated Gangetic basin. At the same time, ISMR has a high cloud cover (CF of 0.72, Figure 1B) and receives widespread rainfall (RF of 9.4 mm, Figure 1C) during monsoon. This implies rapid build up of aerosol concentration over this region after every rainfall event mainly due to high emission rate and geography-induced accumulation of anthropogenic aerosols. Thus, collocation of heavy pollution and

abundant moisture over ISMR makes it an ideal region to investigate aerosol-cloud-rainfall

associations [*Shrestha and Barros*, 2010].

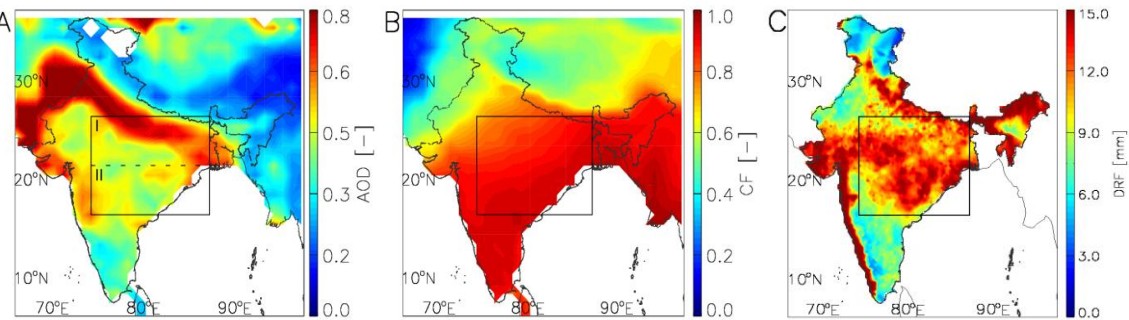

**Figure 1.**Climatological mean of A) aerosol optical depth, B)cloud fraction and C) daily
rainfall for June through September 2002-2013. The black square box indicates the Indian
summer monsoon region (ISMR) focussed in our analysis. Panel A illustrate the boundaries
of regions I and II, used for sub analysis (see the text).

**2.2. Analysis of aerosol impact on cloud radiative forcing**

Clouds increase Earth's albedo and cool the atmosphere by reflecting solar radiation

to space as well as warm the atmosphere by absorbing Earth's outgoing longwave radiation

[*Trenberth et al.*, 2009]. Thus, aerosol microphysical effect in convective clouds will

manifest itself in association between cloud radiative forcing and aerosol [*Feingold et al.*,

2016; *Koren et al.*, 2010b]. Here, the Clouds and the Earth's Radiant Energy System

(CERES) [*Wielicki et al.*, 1996] retrieved outgoing shortwave (SW) and longwave (LW)

radiation at top-of-the-atmosphere (TOA) was also used to illustrate the aerosol-induced

changes to cloud radiative forcing. The CERES fluxes were sorted and averaged as a function

of AOD (similar to correlation analysis detailed in Section 2.1) for two different scenarios,

i.e. all sky and clear sky. While aerosol radiative forcing during clear sky scenario includes

only aerosol direct effect, radiative forcing due to aerosol indirect effect can be estimated

from the net difference between all sky and clear sky scenario.

### 2.3. Analysis of aerosol impact on liquid- and ice-phase cloud microphysics

MODIS observations of cloud top liquid effective radius ($R_e$)as a function of cloud top pressure for convective cloud fields can be assumed as a composite $R_e$-altitude profile obtained from tracking the space-time evolution of individual clouds [*Lensky and Rosenfeld*, 2006]. Insensitivity of $R_e$ to spatial variations at any particular altitude is also reported during CAIPEEX campaign over ISMR [*Prabha et al.*, 2011]. CTP and $R_e$ was segregated into groups of low (AOD<33 percentiles) and high (AOD>67 percentiles) aerosol loading regime using collocated AOD values. $R_e$ as a function of CTP was compared between low and high aerosol regimes. The aerosol associated differences in growth of cloud droplets with height from these CTP- $R_e$ profiles were used to infer aerosol-induced differences in warm cloud microphysical processes and the initiation of rain over ISMR [*Rosenfeld et al.*, 2014 and references therein].

CloudSat-retrieved profiles of liquid- phase and ice-phase water content as well as ice-phase effective radius ($R_{e, ICE}$) available at 75 meters vertical resolution within ISMR [*Austin et al.*, 2009; *Stephens et al.*, 2002] were also segregated intolow (AOD<33 percentiles) and high (AOD>67 percentiles) aerosol loading conditions. The mean microphysical variables along with their variability(profiles indicating $25^{th}$ and $75^{th}$ percentile) for low and high aerosol bins were plotted against altitude to visualize the net increase or decrease in liquid-phase water content, ice-phase water content and size of ice-phase hydrometeors at different altitudes with increase in aerosol loading. The (two sample) Student's *t*-test was used for statistical hypothesis testing about mean of the groups in each subplot.

### 2.4. Modeling aerosol-cloud-rainfall associations: A case study of a heavy rainfall event over ISMR

The WRF model is a regional numerical weather prediction system principally developed by the National Centre for Atmospheric Research (NCAR) in collaboration with several research institutions in U.S. The Advanced Research WRF (ARW) version 3.6 along with a newly coupled fast version of spectral bin microphysics (WRF-SBM) is used to perform three idealized supercell simulations of a typical heavy rainfall event over ISMR. The spectral bin microphysics scheme is specially designed to study aerosol effect on cloud microphysics, dynamics, and precipitation based on solving kinetic equations system for size distribution functions described using 33 doubling mass bins [*Khain and Lynn*, 2009; *Khain et al.*, 2004; *Lynn and Khain*, 2007]. In fast SBM four size distributions are solved, one each for CCN, water drops, low density ice particles and high density ice particles. All ice crystals (sizes<150μm) and snow (sizes<150 μm) are calculated in the low density ice particle size distribution. Graupel and hail are grouped to the high-density ice, represented with one size distribution without separation. The empirical dependence $N = N_{o*}S^{k}$ is used to calculated the initial (at time, t = 0) CCN size distribution [see *Khain et al.*, 2000 for details]; where, $N_o$ and $k$ are parameters which varies with aerosol number concentration and chemical composition, respectively and $N$ is the concentration of nucleated droplets at supersaturation, $S$, (in %) with respect to water. At each time step the critical aerosol activation diameter of cloud droplets is calculated from the value of $S$ (using Kohler theory). It explicitly calculates nucleation of droplets and ice crystals, droplet freezing, condensation, coalescence growth, deposition growth, evaporation, sublimation, riming, melting and breakup of the categorized hydrometeor particles. Details about the parameterizations used for these processes can be found in previous studies [*Khain and Lynn*, 2009; *Khain et al.*, 2004; *Lynn and Khain*, 2007].

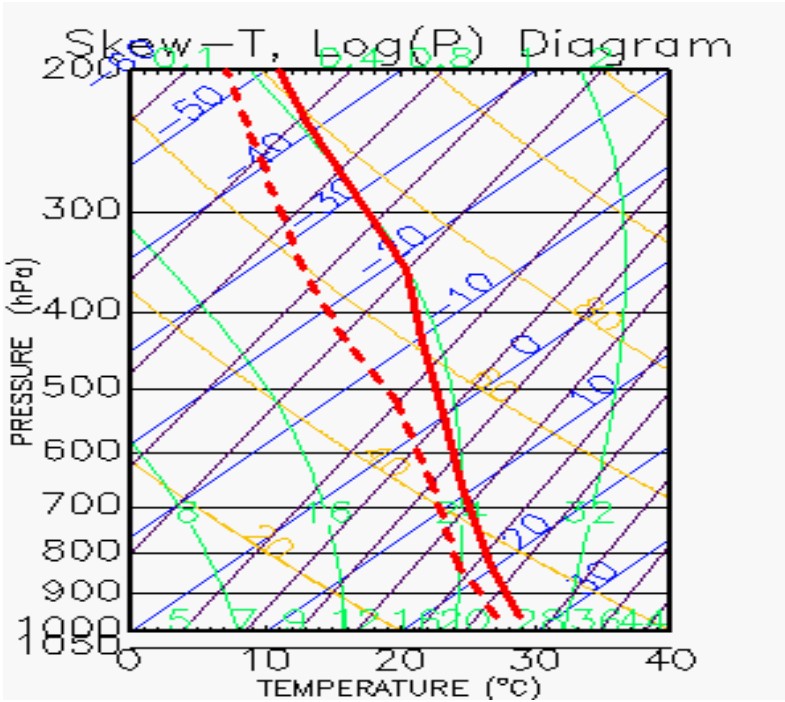

**Figure 2**.Skew-T - log-P diagram illustrating the initial conditions of dew point temperature (red hashed line) and atmospheric temperature (red solid line) used in all the three WRF-SBM idealized simulations. Blue, yellow, green, black and purple lines indicate lines of constant temperature (isotherm), potential temperature and equivalent potential temperature, pressure (isobar), and saturation mixing ratio, respectively.

We found that the mean relative humidity in lower troposphere remains high over ISMR during moderate and heavy rainfall events (RF>6 mm) using GDAS data (Figure not shown). Our simulations were initiated with morning Radiosonde measurements (on 23[rd] August 2009) from Patna station of Indian Meteorological Department (Figure 2). A mesoscale convective system was prevalent over Patna during 22-25 August 2009. The moisture mixing ratio was high in lower troposphere during this event (Figure 2) which is typical of moderate to heavy rainfall event over ISMR. This particular period was selected because measurements of CCN spectrum near cloud base were also available from CAIPEEX campaign over the region [*Prabha et al.*, 2012]. We performed three simulations with same initial thermodynamic conditions but different initial $N_o$ to represent low ($N_o = 4500$ particles/cm$^3$), medium ($N_o = 9000$ particles/cm$^3$) and heavy ($N_o = 15500$ particles/cm$^3$) aerosol loading conditions, hereafter referred to as Ex1, Ex2 and Ex3, respectively. The

simulations were performed for 160 minutes at a resolution of 1 km over a domain of 300 km

x 300 km. The number of vertical sigma levels was 41 and the top height was about 20 km.

Rayleigh damping was used to damp the fluctuations reaching the upper troposphere in the

idealized simulation[*Khain et al.*, 2005]. An exponentially decreasing (both horizontally and

vertically) temperature pulse of 3°C was used to trigger the storm[*Khain and Lynn*, 2009;

*Khain et al.*, 2004; *Lynn and Khain*, 2007]. A comparison of droplet size distribution,

microphysical profiles, vertical velocity, column accumulated water content of various cloud

species and surface rainfall from these simulations illustrate the process level linkage

between aerosol increase and surface rainfall. The simulation output of mass size

distributions of water droplets, low density ice particles and high density ice particles were

recorded every 15 minutes of model time. Assuming that all the hydrometeors were spherical

shaped, we calculated the number-size distribution from the mass-size distribution by using

the bulk radius-density functions specified in SBM for each hydrometeor (shown in Figure 1

of [*Iguchi et al.*, 2012]). The bulk effective radius ($R_e$) of each size distribution was

calculated as shown in Equation 1 below;

$$R_e = \frac{\sum_{i=3}^{33} r_i^3 N_i}{\sum_{i=3}^{33} r_i^2 N_i} \qquad (1)$$

Where, $r_i$ is a half of the maximum diameter and $N_i$ is the particle number concentrations of

$i^{th}$ bin. For calculating $R_e$ of cloud droplets the bins with diameter <50 $\mu m$ was considered.

We used $1^{st}$ -$17^{th}$ bins and$17^{th}$-$33^{th}$ bins of low density ice hydrometeors size distribution,

separately, to calculate $R_{e,\ ice}$ and $R_{e,\ snow}$ respectively. $R_{e,\ graupel}$ was calculated using size

distribution of high density ice particles.

### 2.5. Analysis of possible caveats in correlation analysis

      It is well documented that the aerosol-cloud correlation analysis using satellite data

can be affected by one or more of the following factors: (1) positive correlation of variability

in aerosol and cloud-rainfall fields with meteorological variations, which are the true modifiers of cloud and rainfall properties [*Chakraborty et al.*, 2016; *Kourtidis et al.*, 2015; *Ten Hoeve et al.*, 2011] and (2) cloud contamination of retrieved AOD values due to aerosol humidification effect [*Boucher and Quaas*, 2013; *Gryspeerdt et al.*, 2014]. (3) Inaccurate representation of wet scavenging effect in satellite retrieved AOD dataset [*Grandey et al.*, 2013; *Grandey et al.*, 2014; *Yang et al.*, 2016].Therefore, we have critically investigated the plausible role of these factors in our analyses as presented below.

### 2.5.1.Influence of meteorological variability

Here, we obtained various meteorological fields from the NOAA-NCEP Global Data Assimilation System (GDAS) dataset [*Parrish and Derber*, 1992] as an approximation for the meteorological conditions at the same time and location of the satellite observations. Since the NOAA-NCEP GDAS assimilated product does not contain direct information on the aerosol microphysical effects, it is a suitable tool to investigate if the meteorological variations favoured aerosol accumulation under wet/cloudy conditions[*Koren et al.*, 2010a]. GDAS variables at $1^o$ spatial resolution and 21 vertical model levels (1000 hPa - 100 hPa) over ISMR from the 12:00 LT run were used. First, correlation of different GDAS meteorological variables with cloud fraction, daily rainfall and AOD, separately, using all grid points within ISMR at each model vertical level was performed. Based on the correlation analysis,the likely meteorological variables (with correlation coefficient > 0.25) which can affect cloud and rainfall properties in ISMR were identified. Next, we made narrow regimes of these key meteorological variables to constrain the variability in these meteorological factors and repeated the correlation analysis of AOD-cloud-rainfall gradients. This approach can be assumed to be similar to simulating the effect of increasing aerosol loading on cloud-rainfall system for similar meteorology

### 2.5.2. Cloud contamination of aerosol retrievals

Aerosol-cloud-rainfall studies based on satellite data are, in part, biased by aerosol humidification effect due to uncertainties in retrieved AOD from near cloud pixels. For instance, an increase in surface area of aerosol due to water uptake may cause elevated AOD levels measured in the vicinity of clouds [*Boucher and Quaas*, 2013]. The humidification effect on the AOD depends on the variability range of ambient RH [*O Altaratz et al.*, 2013]. Here, we used radiosonde measurements (JJAS, 2002 to 2013) from World Metrological Organization stations [*Durre et al.*, 2006], within ISMR (Table 2)to identify profiles that had potential of cloud formation. Specifically, the selected profiles had unstable layer below lifting condensation level (LCL). However, the profiles suggesting low level clouds (mean RH below LCL>98%) were removed. A major portion of aerosols contributing to columnar AOD are usually present below 3 km altitude over ISMR during monsoon/cloudy conditions. Thus, we focused this analysis for RH below 3 km altitude. Also, the changes in mean RH values associated with the change in cloud vertical extent was calculated based on O Altaratz et al., (2013). The height above the level of free convection where the theoretical temperature of a buoyantly rising moist parcel (following wet adiabatic lapse rate) becomes equal to the temperature of the environment is referred to as equilibrium level. The height of atmospheric layer between LCL and the equilibrium level is referred to as the cloudy layer height (CLH). Also, in case of the presence of inversion layer, the top of the CLH is determined as the base of the lowest inversion layer located above the LCL. Based on median CLH, the selected profiles at each station were divided into two subsets of equal number of samples representing shallower and deeper clouds. The bias in mean RH between shallower and deeper clouds for each station was calculated to illustrate the influence of cloud height on the RH variability.

*Bar-Or et al.* [2012] have parameterized RH in cloudy atmosphere as a function of the distance from the nearest cloud edges. Given the hygroscopic parameter, *k,* this parameterization can be used to simulate hygroscopic properties and model the humidified aerosol optical depth. *Bhattu and Tripathi* [2014] have reported that *k* of ambient aerosol over Kanpur (in Gangetic basin ) during monsoon is 0.14±0.06.  Accordingly, we have considered minimum (maximum) *k* over ISMR as 0.1 (0.2), and have used the parameterization to estimate the change in AOD due to the observed variation in RH field. First, the range in RH variation was scaled as distance from the nearest cloud (using Figure 3 of Bar-Or *et al.* 2012) and then the change in AOD was estimated (using Figure 6 of Bar-Or *et al.* 2012) for each subset.

**2.5.3. Effect of under-representation of wet scavenging effect on retrieved AOD values**

Aerosols present below cloudy pixels are not visible to satellite. To circumvent this limitation in investigating aerosol-cloud-rainfall association, it can be reasonable to assum that the mean aerosol distribution below the non-raining cloudy pixels is similar in magnitude to the aerosol distribution of the non-cloudy pixels within a 1$^o$ x 1$^o$ grid box. Nevertheless, aerosols below cloudy pixels, where rainfall occurs, are subject to depletion due to wet scavenging effect. Thus, wet scavenging effect might not be accurately represented in the MODIS retrieved AOD dataset used in our study. Modelling studies suggest that this artifact in the satellite retrieved AOD values can significantly affect the magnitude as well as the sign of the aerosol-cloud-rainfall associations [*Grandey et al.*, 2013; *Grandey et al.*, 2014; *Yang et al.*, 2016]. At the same time, *Gryspeerdt et al.*, (2015)  [*Gryspeerdt et al.*, 2015] have recently illustrated that the aerosol in neighbouring cloud-free regions may be more representative for aerosol-cloud interaction studies than the below-cloud aerosol using a high resolution regional model, justifying the methodology used in their study. The main limitation in investigating the impact of probable inaccuracy in representing  wet scavenging effect on our

analysis is lack of collocated measurements of aerosol-cloud-rainfall at temporal resolution of rainfall events from space-borne measurements. Hence, we used collocated hourly measurements of aerosol and rainfall over Indian Institute of Technology, Kanpur (IITK) as a representative case study dataset to investigate the possible effect of wet scavenging on

aerosol-rainfall associations within ISMR.

AErosol RObotic NETwork (AERONET), is a global network of ground based remote sensing stations that provides quality-controlled measurements of aerosol optical depth with high accuracy [*Dubovik and King*, 2000; *Holben et al.*, 1998]. Hourly averages of AOD (550 nm) used in this analysis were obtained from the quality ensured Level-2 product

of AERONET site deployed in the IITK campus. Rainfall events were identified from collocated rain gauge measurements near AERONET station within IITK campus between April-October; 2006-2015. We have also included the months of April, May and October to increase the number of sample points. Rainfall amount of all the rainfall events were sorted as a function of collocated AERONET-AOD values (mean of AERONET-AOD measurements

within ± 4 hour of the start/end of the rainfall) into 5 equal bins of 20 percentiles each. As AERONET-AOD measurements were available only between sunrise and sunset, we have used AOD values of late evening measurements as representative of aerosol loading during the first rainfall event (if any) at night-time. However, in case of more than one rainfall events at night, only the first rainfall event is considered in this analysis. Nearly half of the

AOD-rainfall samples used here included AOD measurements within 4 hours after the end of any rainfall event, and therefore, this includes a wet scavenging effect of rainfall on AOD measurements. To reproduce another specific scenario, only the rainfall-AOD samples when AOD measurement was available before start of rainfall events were collected and sorted as a function of AOD into 5 equal bins of 20 percentiles each. This restricted sampling does not

include the wet scavenging effect as only the AOD-values before the start of rainfall in each

rainfall event were used. The average of rainfall amount for each bin was plotted against mean AOD values under both scenarios to illustrate the difference in aerosol-rainfall association due to exclusion of wet scavenging effect within ISMR.

**3. Results and Discussion**

**3.1.Cloud, rainfall and radiation associations with aerosol loading**

Figure 3A shows the relationship between AOD and IMD RF. RF increased from 5.9 mm to 7.1 mm as AOD increased from 0.25 to 0.75. A similar relationship was also observed in case of TRMM PR in Figure 3B. Precipitation rate increased from 0.31 mm/hr to 0.38 mm/hr for the same amount of increase in AOD (0.25 to 0.75). Concurrent analysis of aerosol and cloud properties showed aerosol-induced modifications in cloud macrophysics. Widening of clouds was observed as cloud fraction increased from 0.78 to 0.92 with increase in AOD from 0.25 to 0.75 (Figure 3C). A monotonic decrease in CTP and CTT (Figures 3D and 3E), nearly by 200 hPa and $22^o$ K, respectively, for the same increment in AOD, further indicate vertical deepening of the cloud with increasing aerosol loading. Aerosol-cloud studies have reported reduction in cloudiness under high AOD for regions with high absorbing aerosol loading[*Koren et al.*, 2004; *Small et al.*, 2011]. Widespread cloud coverage over ISMR (CF of ~0.75 for AOD ~0.3 in Figure 3) induces substantial reduction in the incoming solar radiation [*Padma Kumari and Goswami*, 2010], which may result in reduced interaction between absorbing aerosols and shortwave radiation. This explains that, despite the high emission rate of absorbing aerosols over ISMR [*Bond et al.*, 2004], the aerosol-induced cloud inhibition effect seemed to have been reduced to a second order process during Indian summer monsoon. For a sanity check, we have re-analyzed cloud and rainfall associations with aerosol loading by dividing ISMR into two sub-regions (shown in Figure 1A). A similar aerosol-cloud-rainfall associations (in both the regions) were observed to that seen in Figure

3. In addition, the analysis was also repeated by segregating the dataset into low level

(850hPa>CTP>500 hPa) and high level clouds (CTP<500 hPa) (Figure not shown). Despite

the considerable differences in mean CTP and CTT found between low- and high-level

clouds, the general associations was similar in both the regimes (as in Figure 3). Analysis of

individual months viz; June, July, August and September also illustrated similar positive

associations as seen in Figure 3 indicating negligible intra-seasonality in the observed

associations.

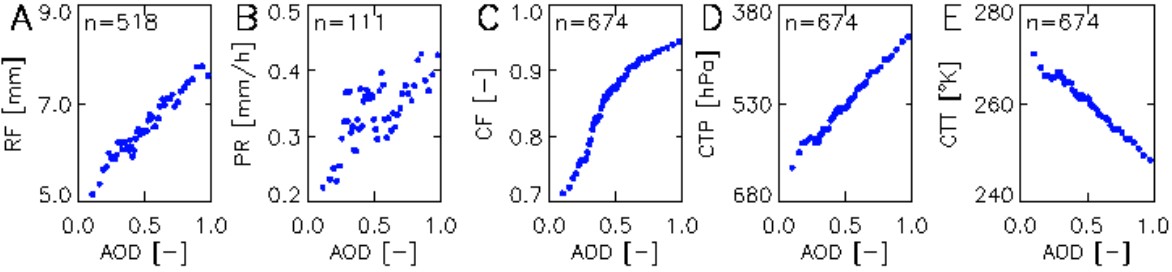

**Figure 3.** Associations of (A) daily rainfall, (B) precipitation rate, (C) cloud fraction, (D)
cloud top pressure, and (E) cloud top temperature with AOD. The collocated data points for
these five variables (A-E) were sorted as a function of AOD over ISMR during JJAS 2002-
2013. The total number of collocated data points (50×n) are then used to create 50 AOD bins
of 'n' number of samples (2 percentile) each. Each scatter point is the average of these equal
'n' numbers of data points mentioned in each respective panels.

      The observed associations in Figure 3 are in line with the recent aerosol-cloud-rainfall

association studies under continental conditions[*Kourtidis et al.*, 2015; *Myhre et al.*, 2007;

*Ten Hoeve et al.*, 2011], CTP[*Li et al.*, 2011; *Myhre et al.*, 2007; *Yan et al.*, 2014] and rainfall

[*Gonçalves et al.*, 2015; *Heiblum et al.*, 2012]. These studies suggested that aerosol-induced

changes in cloud dynamics and microphysics are the potential causal mechanism for the

aerosol-cloud-rainfall linear dependence. Over Indian region, previous studies have compared

MODIS-observed cloud microphysical properties between low and high aerosol loading to

demonstrate aerosol microphysical effect and its linkage to inter-annual variations in seasonal

rainfall [*Abish and Mohanakumar*, 2011; *Panicker et al.*, 2010; *Ramachandran and Kedia*,

2013]. Aerosol impacts on cloud microphysics over central India based on ground based

measurements is also evident[*Harikishan et al.*, 2016; *Tripathi et al.*, 2007]. Aircraft measurements during Cloud Aerosol Interaction and Precipitation Enhancement EXperiment (CAIPEEX) campaign over ISMR have provided unprecedented evidence of aerosol microphysical effect on cloud droplet distribution and warm rainfall suppression over

ISMR[*Konwar et al.*, 2012; *Pandithurai et al.*, 2012; *Prabha et al.*, 2011]. Recently,*Sengupta et al.* [2013]have also discussed the possible aerosol-induced deepening of clouds with evolution of Indian monsoon using MODIS retrieved CTP.

Next, aerosol-related convective invigoration was investigated using CERES retrieved outgoing radiative fluxes at the top of the atmosphere. Our analyzes showed that for every

unit increase in AOD, reflected SW radiation increased by ~68 W/m$^2$, whereas LW decreased by ~26 W/m$^2$ at the top of the atmosphere for all sky scenario (Figure 4A). Taller clouds exhibit colder cloud tops as they are in a thermodynamic balance with the environment, therefore, the observed decrease in LW with increase in AOD further provides evidence of aerosol-induced cloud invigoration over ISMR [*Koren et al.*, 2014; *Koren et al.*, 2010b].

Increased cloudiness was also evidenced as the cloud albedo increased, thereby reflecting back more SW radiation at the top of the atmosphere. A large number of small ice crystals formed in the upper troposphere due to cloud invigoration eventually get aligned as larger and longer-lived anvils detrained from cloud tops[*Fan et al.*, 2013]. Such  anvil expansion effect of aerosol [*Rosenfeld et al.*, 2014] may also contribute to the aerosol-associated

increase in  SW radiative forcing. Quantitatively, the net cooling per unit increase in AOD (Figure 4B) under clear sky scenario was ~13 W/m$^2$,whereas the net cooling for same change in AOD under cloudy condition was twice more than that under clear sky scenario i.e. ~30 W/m$^2$.

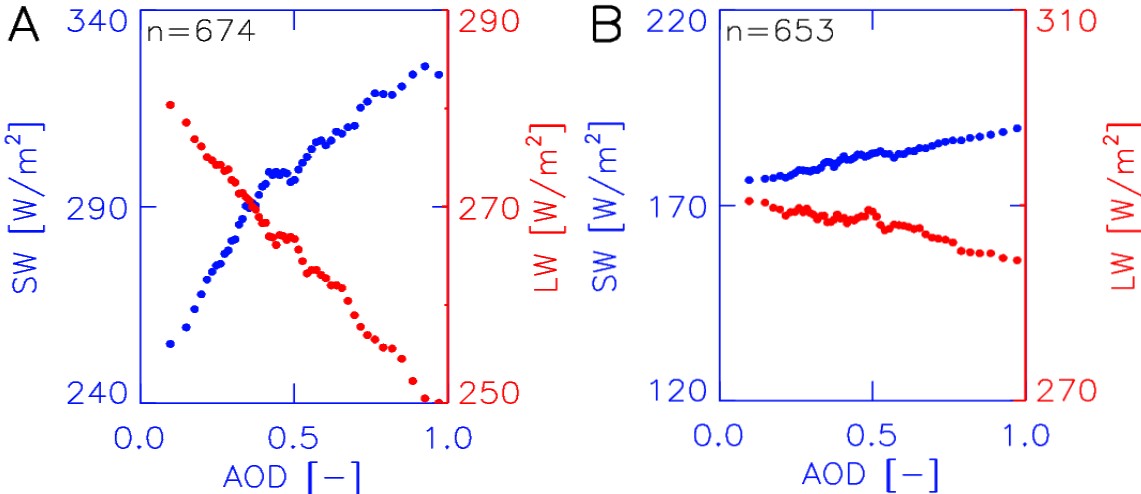

**Figure 4**.Association of CERES retrieved incoming shortwave (SW) and outgoing longwave (LW) radiation with AOD for (A) all-sky and (B) clear-sky scenario over ISMR during JJAS 2002-2013. The collocated data points for both SW and LW as a function of AOD were first sorted. The total number of collocated data points (50×n) are then used to create 50 AOD bins of 'n' number of samples (2 percentile) each. Each scatter point is the average of these equal 'n' numbers of data points mentioned in each respective panels.

### 3.2Aerosol-induced cloud invigoration

### 3.2.1,Effect of aerosol-related changes in  microphysical processes

Many studies have shown that the onset of warm rain and collision-coalescence process are dependent on the CCN concentration [*Freud et al.*, 2011 and references therein]. MODIS retrieved droplet effective radius as a function of CTP grouped under low and high aerosol loading cases can be used to investigate the aerosol-induced differences in warm rain processes like diffusion and coalescence processes[*Rosenfeld et al.*, 2014]. In Figure 5, we present cloud microphysical changes for low and high aerosol loading using MODIS and CLOUDSAT datasets.

Figure 5A illustrates that $R_e$ of liquid droplets near cloud base was smaller (6 μm) in clouds developed under higher AOD conditions which is in agreement with aerosol first indirect effect [*Twomey*, 1974]. In addition, the vertical growth of  $R_e$ under polluted conditions increased at a gradual rate (~3 μm/100 hPa) for $R_e$<14 μm compared to the vertical gradient of increase in $R_e$ (~10 μm/100 hPa) in relatively clean clouds (low aerosol

loading). Also note that the altitude difference between cloud base and onset of warm rain was smaller under low AOD cases (~50 hPa) compared to that at high AOD cases (~250 hPa). Concurrently, the mean $R_e$ for high AOD cases was very small (~10 μm) near the freezing level compared to low AOD indicating increase in droplets of smaller size at higher

levels with increase in aerosol loading (Figure 5A). Thus, significant increase and sustenance of smaller supercooled liquid drops was found above freezing level under polluted conditions. Aircraft measurement of clouds developed under dirty conditions during CAIPEEX campaign over ISMR have also documented that $R_e$ remained below 14 μm up to 500 hPa altitude and formation of rain drops mainly initiated as supercooled raindrops at ~

400 hPa [*Konwar et al.*, 2012; *Prabha et al.*, 2011].

From CLOUDSAT analyses, mean ice-phase effective radius ($R_{e,ICE}$) for high aerosol loading was found to be 8-10% greater (significant at >95% confidence interval) throughout the cloud layer compared to that for low aerosol loading at the same altitude (Figure 5B), indicative of the formation of bigger sized ice-phase hydrometeors under high

aerosol loading. Figure 5C shows the difference (high aerosol - low aerosol) in mean liquid-phase and ice-phase water content. Significant enhancements in ice-phase water content was clearly evident under high aerosol loading (Figure 5C). The increase in mass concentration of ice-phase hydrometeors was ~50 mg/m$^3$ at altitudes 8-13 km. Similar increase in number concentration of ice hydrometeors was also observed from CLOUDSAT observations (figure

not shown).

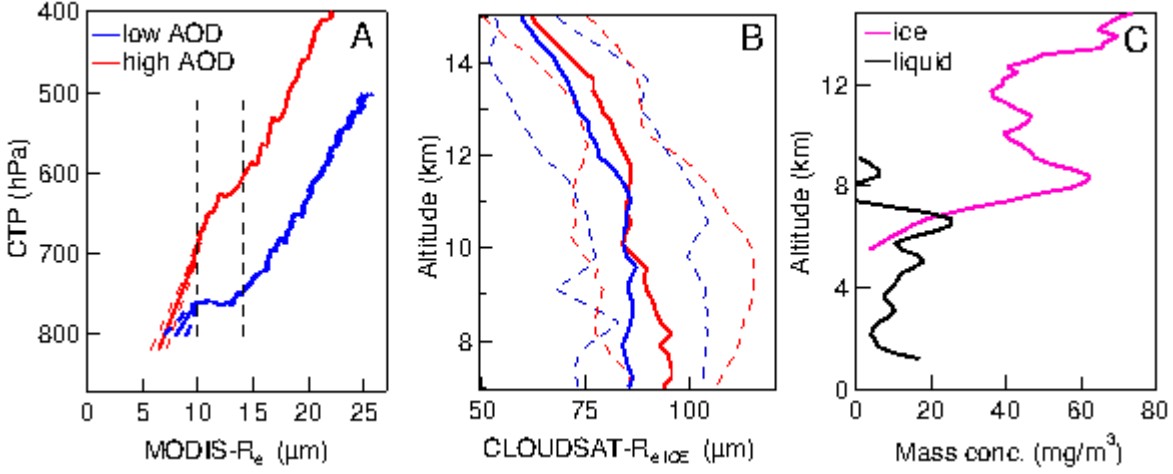

**Figure 5**.Observed differences in cloud microphysical propertiesfor low and high aerosol loadings cases. A) MODIS observed mean profiles of liquid-phase effective radius ($R_e$), B) CLOUDSAT observed mean profiles of ice-phase effective radius ($R_{e, ICE}$) under low (blue) and high (red) aerosol loading conditions. The dotted lines represent $25^{th}$ and $75^{th}$ percentiles, respectively. C) Difference (high AOD - low AOD) in mean profiles of liquid-phase (black) and ice-phase (pink) water content as observed from CLOUDSAT.

### 3.2.2.Modelling aerosol microphysical effect for a typical rainfall event during ISM

In order to further investigate the process level insights to our observational findings, we conducted model simulations using WRF-SBM for a typical mesoscale convective system over ISMR. Three idealized supercell simulations (Ex1, Ex2 and Ex3 as explained above) were performed with the observed CCN spectra being lowest for Ex1 and highest for Ex3.

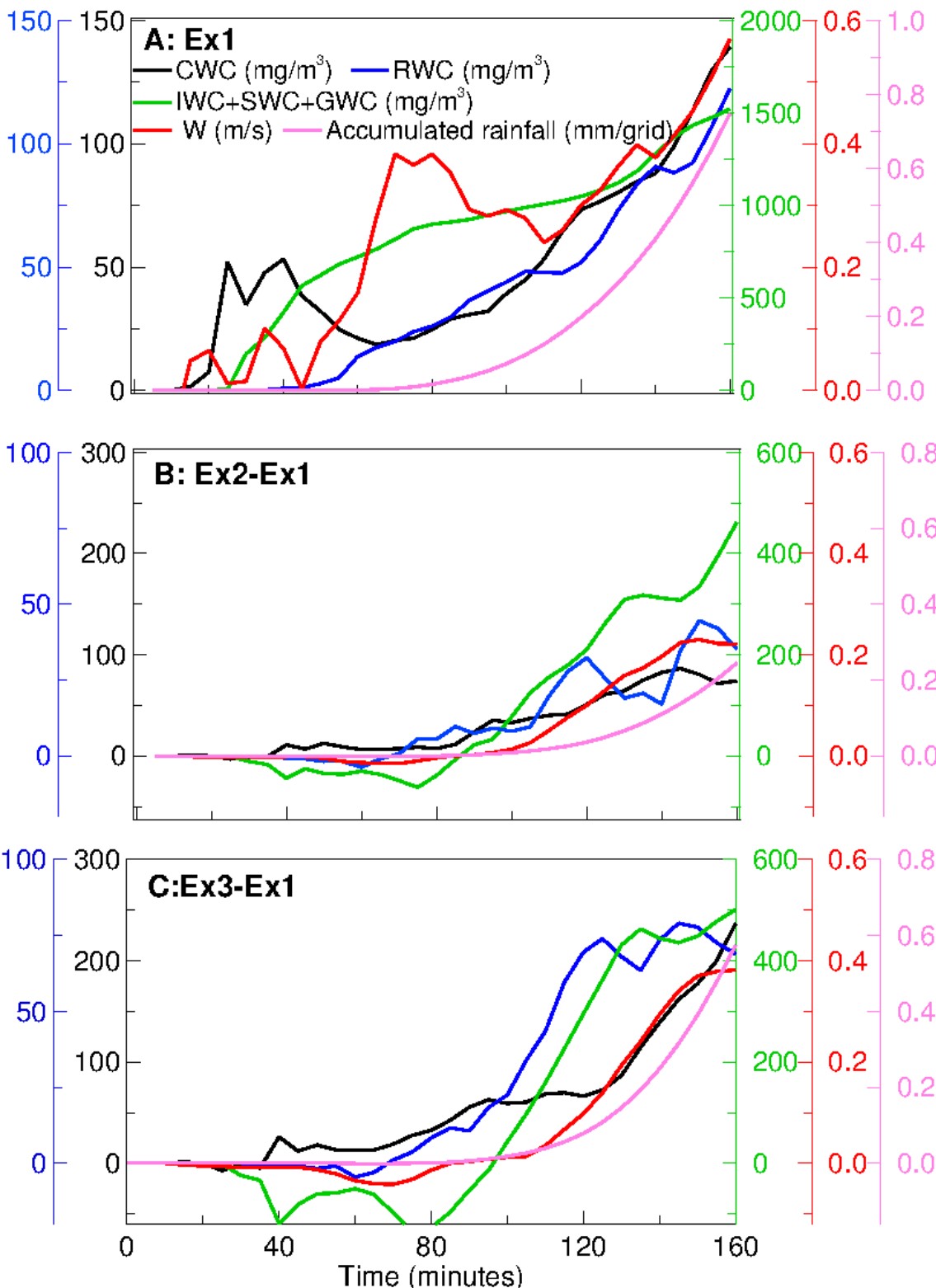

**Figure 6.** A)Time evolution of column integrated domain averaged cloud water content (CWC; black), rain water content (RWC; blue), summation of ice water content, graupel water content and snow water content (IWC+GWC+SWC; green), vertical velocity (red) and accumulated surface rainfall (pink) for simulation Ex1. B) Same as Panel A, but for simulated differences between Ex2 and Ex1. C) Same as Panel A, but for simulated differences between Ex3 and Ex1.

Figure 6A shows the time evolution of domain averaged mean columnar cloud

water content (CWC), rain water content (RWC), summation of ice phase hydrometeors i.e.

snow, graupel and ice water content (SWC+GWC+IWC), vertical velocity (W), and

accumulated rainfall for low CCN (aerosol) condition. It can be seen that convection was

strong after 50 minutes (consistent updrafts > 0.2 m/s), with corresponding enhancements

in CWC, RWC and hydrometeors till the end of simulation. The domain-averaged

accumulated rainfall was found to be ~0.8 mm/grid at the end of simulation. The simulated

differences between high CCN and low CCN conditions (Figures 6B and 6C)clearly show

significant intensification in the microphysical and dynamic variables with increase in CCN

concentration. The magnitude of W, CWC, RWC and ice-phase water content increased in

both simulations (Ex2 and Ex3), as compared to simulation Ex1. The simultaneous increase

in accumulated rainfall was also evident with increase in CCN concentrations, mainly

during the last half of the simulations. The estimated AOD for prescribed CCN scenarios in

Ex1, Ex2 and Ex3 at 0.4 % supersaturation are 0.42, 0.62 and 0.91, respectively (using

empirical formula given by *Andreae et al., [2009]*). The observed increase in accumulated

rainfall was found to be 0.68 mm and 0.28mm for an increase in AOD of 0.5 (Ex3-Ex1) and

0.3 (Ex2-Ex1), respectively, suggesting a nearly linear relationship in CCN-cloud-rainfall

association as observed in Figure 2. Nevertheless, a closer look at Figures 6B and 6C reveal

a temporal delay in initial formation of RWC, ice-phase hydrometeors and surface rainfall

with increase in CCN concentrations. This can be understood from the negative values of

differences in RWC, total water content of ice-phase hydrometeors and rainfall between 40-

100 minutes of simulation. However, the increase in rainfall amount with increase in CCN

concentration in later stage of simulation was manifold compared to the initial suppression

of warm rainfall eventually leading to the enhancement of accumulated rainfall throughout

25   the storm domain(Figure not shown).

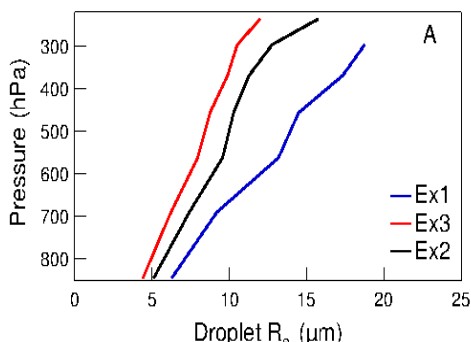

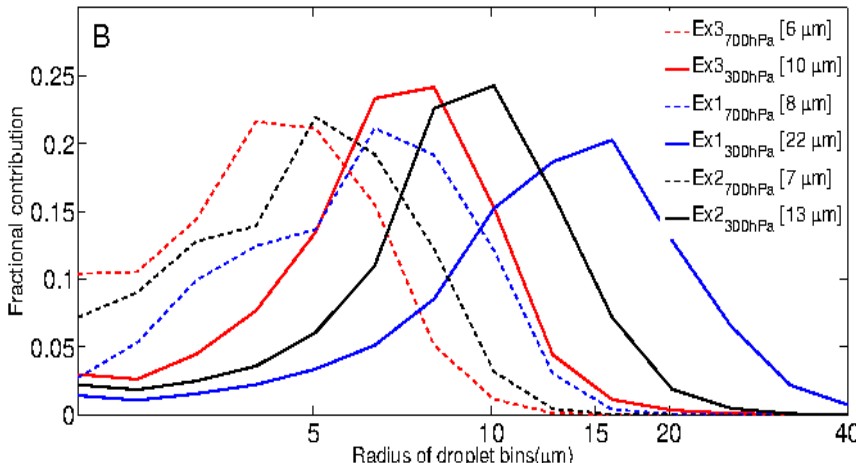

Figure 7: A) mean droplet $R_e$ versus CTP for low (Ex1; blue), medium (Ex2; black) and high (Ex3; red) CCN scenario. B) Droplet size distribution spectra of Ex1 (blue), Ex2 (black) and Ex3 (red) simulations at 700 hPa (dashed lines) and 300 hPa (solid lines). The corresponding effective radius values are mentioned in the legends in square brackets. Fractional contribution is calculated by dividing the mass concentration of each bin with the total mass concentration.

Figure 7A illustrates the simulated time and domain averaged profiles of droplet

effective radius for Ex1 and Ex3. It can be seen that droplet $R_e$ in Ex3 simulation was lower

compared to that of Ex1 throughout the cloud column and the differences increased with

altitudes, indicative of the slower growth of cloud droplets for high CCN condition (Ex3) as

compared to low CCN (Ex1), in line with our observation from MODIS analyzes. For

instance, the difference in droplet at 700 hPa and 300 hPa was ~3 µm and ~8 µm,

respectively (Figure 7B). The simulated spectral width of the droplet size distributions for

Ex3 and Ex1 also showed a significant shift of the droplet spectral toward lower $R_e$ with

increase in CCN. It can be seen that increase in CCN concentration also leads to narrowing of droplet spectral at same altitude.

The aerosol-induced increase (Ex3-Ex1) in time and domain averaged CWC, RWC, IWC, SWC, GWC at different altitudes is shown in Figure 8A. Modelling results

also show that the maximum increase in CWC (23 mg/m$^3$) was above freezing level at altitude ~7 km, which suggest that the increase in CCN caused increase in supercooled liquid droplets. Similar plots of mean W and temperature differences averaged over cloudy pixels (Figure 8B) shows considerable increase in temperature and W at altitudes corresponding to increase in CWC (i.e. below 8 km), mainly due to enhanced release of

latent heat of condensation.

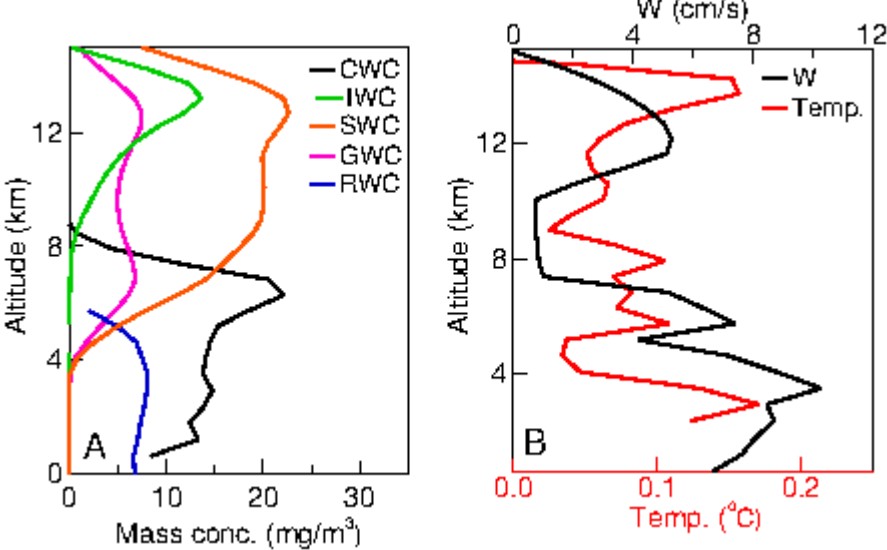

Figure 8. A) Simulated difference (Ex3-Ex1) in mean profiles of cloud water content, rain water content, ice water content, graupel water content, and snow water content. B) CCN induced difference (Ex3-Ex1) in simulated mean profiles of vertical velocity (black) and
temperature (red) for cloudy pixels.

For ice-phase hydrometeors, the majority of the increase was observed in SWC, with a peak above ~12 km altitude. A maxima in the CCN-induced increase in vertical velocity and temperature was also found to be above~12 km (Figure 8). These results

indicate that CCN-induced increase in latent heat of freezing occurred mainly above 12 km,

in turn strengthening the updraft velocity of cloud parcels and hydrometeor formation. Further, snow $R_e$ profiles for simulations Ex1 and Ex3 illustrated that the effective mean radius of snow significantly increased with an increase in CCN concentration between 8 and15 km altitude (Figure 9A). The simulated particle size distribution of snow further explained this behaviour as the mass of particles in bigger sized bins increased in the simulation Ex3 compared to Ex1 (Figure 9B). Similar changes in graupel concentration and particle size distribution for high density ice particles was also found (figure not shown).

It has to be noted that the CCN-induced differences in cloud microphysics and rainfall from this idealized case study simulation should not be directly compared with the decadal scale observational analysis. Moreover, these results are subject to various assumptions and uncertainties within physical parameterizations of the microphysics module used. However, the qualitative similarities in results between the observed aerosol-cloud-rainfall associations and this idealized case study simulation provide confidence in our observational finding that aerosol loading can potentially alter the warm phase and cloud phase microphysics over ISMR. These perturbations are consistent with processes typically associated with aerosol-induced cloud invigoration [*O. Altaratz et al.*, 2014; *Tao et al.*, 2012].

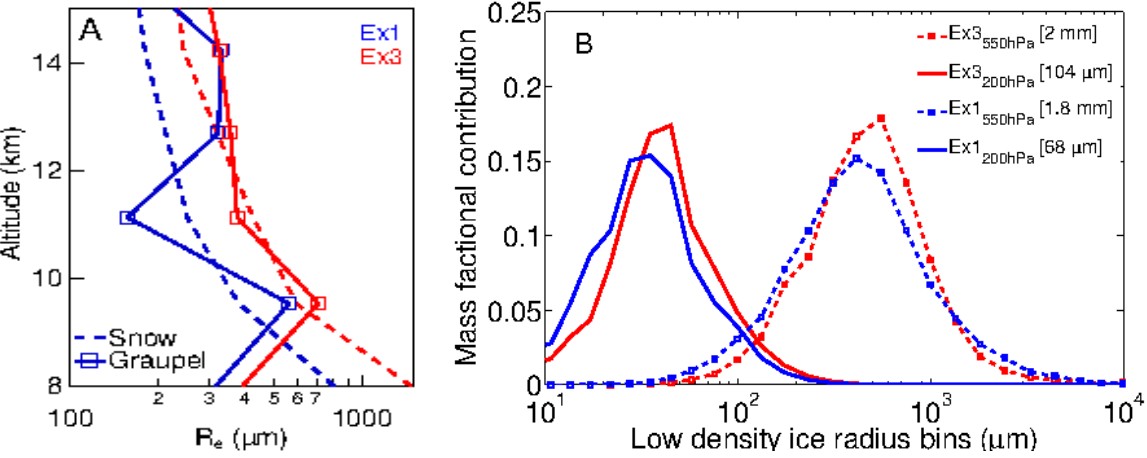

**Figure 9.**A) Simulated mean snow(dashed) and graupel (open square symbol connected by solid line) $R_e$ for low (Ex1, blue) and high (Ex3, red)CCN scenario. B) Simulated size distribution spectra of low density ice particles for Ex1(blue) and Ex3 (red) at 550 hPa (solid lines) and 200 hPa (dashed lines). Fractional contribution is calculated by dividing the mass concentration of each bin with the total mass concentration.

The following chain of processes may explain our observational and/or numerical findings. The growth of cloud droplets near the cloud base is dominated by condensation. However, the growth of droplets near the onset of warm rain ($R_e$ approaches to ~14 μm)is dominated by coalescence [*Rosenfeld et al.*, 2012; *Rosenfeld et al.*, 2014]. The observed

differences in vertical gradient of droplet growth suggest less efficient collision-coalescence process and prolonged condensation process, leading to delayed raindrop formation [*Rosenfeld*, 1999; 2000; *Squires*, 1958; *Warner and Twomey*, 1967]. Such prolonged condensational growth of droplets implies increased condensed water loading, causing more latent heat release and thereby stronger updrafts under higher aerosol loading [*Fan et al.*,

2009; *Khain et al.*, 2005; *Martins et al.*, 2011; *Rosenfeld et al.*, 2008; *van den Heever et al.*, 2011; *Wang*, 2005] . Concurrently, smaller droplet $R_e$ under polluted conditions results in lower effective terminal velocity and higher cloud droplet mobility [*Heiblum et al.*, 2016; *Ilan et al.*, 2015].  Under polluted conditions, then, the aerosol-induced stronger updrafts and enhanced buoyancy would push these smaller condensates above freezing level[*Andreae et*

*al.*, 2004; *Rosenfeld and Lensky*, 1998] which, in turn, would enhance liquid droplets above the freezing level. Nevertheless, the smaller droplets are less efficient in freezing causing delay in the ice-/mix-phase processes which provide sustenance for super-cooled liquid condensates above freezing level [*Rosenfeld and Woodley*, 2000]. These hydrometeors encounter more number of super-cooled liquid droplets while settling from comparatively

higher altitude under gravity. Thus, increased ice-water accretion process [*Ilotoviz et al.*, 2016], increases ice particle $R_e$ under high aerosol loading. Increase in the water mass flux of the smaller droplets at higher altitudes, in principle, releases more latent heat of freezing, and further invigorates the cloud system [*O. Altaratz et al.*, 2014; *Rosenfeld et al.*, 2008]. Such aerosol-induced invigoration also imply the formation of ice-phase hydrometeors at higher

altitudes by freezing of small droplets[*O. Altaratz et al.*, 2014]. Such aerosol-induced

invigorating of clouds ultimately result in wider and deeper clouds, with higher mass concentration of ice-phase hydrometeors, which eventually fall to the surface (Figures 3 and 5)[*Andreae et al.*, 2004; *Koren et al.*, 2005; *Koren et al.*, 2012; *Rosenfeld et al.*, 2008]. Thus, the observed increase in daily rainfall with increasing aerosol loading over ISMR (Figure 3) could stem from the observed differences in warm phase dynamics and microphysics, which, plausibly leads to cloud invigoration and thereby enhances mass concentration of mixed-phase hydrometeors.

### 3.3.1 Decoupling the role of meteorology

Observational and modelled evidences of microphysical impact of aerosol over ISMR suggest causality in the observed relationship between aerosol-cloud and rainfall properties(Figure 3). Here, we examined the plausible role of meteorology in our analyzes. Figure 10 shows correlation coefficients of RF, CF and AOD with GDAS meteorological variables. The meteorological conditions favourable for deeper clouds and heavy rainfall were found to be associated with reduction in AOD (Figure 10). As expected, a positive correlation of CF and RF was observed with relative humidity. However, increase in RH was negatively correlated with aerosol loading, suggesting that cloudy/wet conditions were associated with the reduction in aerosol loading. While CF and RF was found to be negatively correlated with geopotential height (mainly below 500 hPa), AOD was linearly correlated. This suggests that the formation of low pressure zone / presence of high RH at lower atmosphere was favourable for cloud development and rain, but not for aerosol accumulation. These features are consistent with that of heavy rainfall periods of ISM, where, the presence of low pressure zone over ISMR (commonly known as monsoon depressions) is associated for advection of more moisture at lower altitudes, more cloud condensation and occurrence of more rainfall. A recent modelling study has also shown that the propagation of

low pressure system from Bay of Bengal towards Indian landmass, which, brings moisture and heavy rainfall to the region during monsoon, is also associated with a decrease in aerosol concentration over the region [*Sarangi et al.*, 2015]. The decrease might be a combined effect of ingestion by clouds, wet scavenging and dilution effect of relatively clean moist air masses from the Ocean. Positive correlation of wind speed with CF and RF at altitude above 400 hPa was also associated with reduced AOD (Figure 10). The high wind speed above 350 hPa (Figure 10) appears to provide a shearing effect on the cloud development process. Based on the correlation analysis horizontal wind shear (between 500 hPa and 200 hPa), relative humidity and geopotential height (below 500 hPa) were identified as three key meteorological variables (magnitude of correlation coefficient >0.25) affecting cloud and rainfall properties in ISMR.

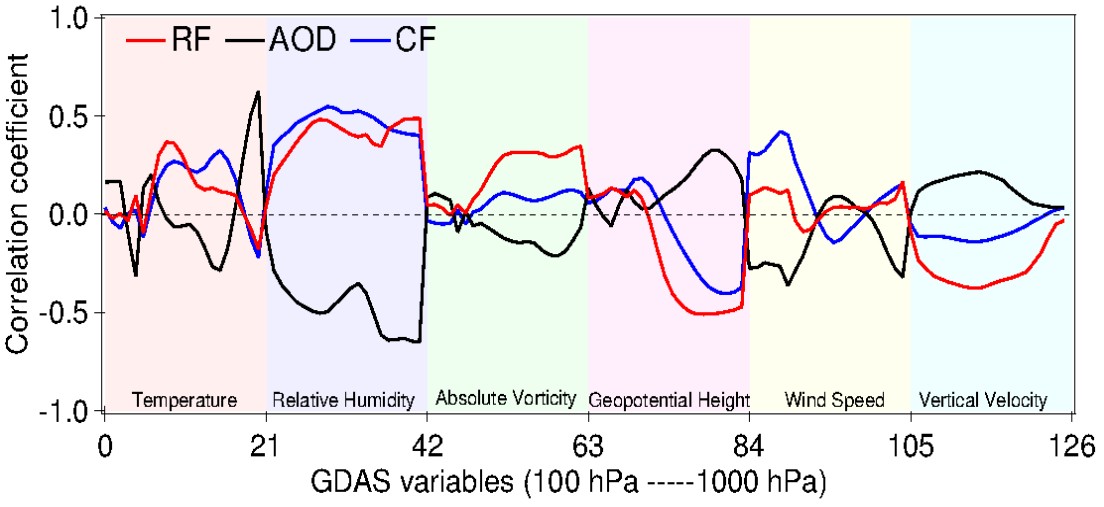

**Figure 10**.Correlation coefficients of accumulated daily rainfall, AOD and cloud fraction with six GDAS meteorological variables over ISMR. Different color shades along the x-axis indicate corresponding meteorological variable and each color shade has 21 divisions which represents corresponding 21 model pressure levels from 100 hPa to1000 hPa (left to right). Correlation analysis was performed at each model pressure level with all collocated samples between two variables (e.g. temperature and RF for x-axis values of 1-21 in case of red color line) over ISMR region for JJAS 2002-2013.

Next, the datasets were segregated into low and high regimes of wind shear, calculated between 200 hPa and 400 hPa, as well as for geopotential height and relative

humidity at 800 hPa pressure level (Figure 11). The low versus high regimes illustrated that

steeper positive gradients in AOD-cloud-RF associations was observed for high relative

humidity and low geo-potential height conditions, but, the magnitude of positive gradient

between RF (and PR)-AOD reduced under high wind shear cases. Spreading of the cloud due

to high wind shear results in hydrometeors falling through relatively drier atmosphere making

smaller droplets (in polluted condition) are more susceptible to evaporation [*Fan et al.*,

2009], thereby, the reduction in PR and RF. Thus, an orthogonal meteorological impact

[*Koren et al.*, 2010a; *Koren et al.*, 2014] was evident on gradients of AOD-cloud-rainfall

associations over ISMR, where, the y-intercept indicates the meteorology effect and the slope

of correlation represents aerosol effect. We have also considered the combined effect of all

the three key meteorological variables by dividing the datasets into 8 regimes (alternate

combination of higher and lower bins of RH, WS and GPH). Our analysis illustrated (Figure

not shown) similar results as seen in Figure 11; positive aerosol-cloud-rainfall association

was evident in all the 8 sub-regimes along with distinct orthogonal effect of ambient

meteorological conditions.

Ground based remote sensing, satellite observations, aircraft measurements and

modelling studies have documented that aerosols are mainly located within the boundary

layer during monsoon period over ISMR [*Mishra and Shibata*, 2012; *Misra et al.*, 2012;

*Sarangi et al.*, 2015]. But, some recent studies have reported that transport of near surface

aerosols to the free troposphere by mesoscale convection results in upper-level accumulation

during summer monsoon, termed as Asian tropopause aerosol layer [*Chakraborty et al.*,

2015; *Vernier et al.*, 2015]. Therefore, another possible pathway through which

meteorological co-variability can influence our correlation analysis over ISMR is due to the

positive association between magnitude of Asian tropopause aerosol layer and AOD.

However, the tropopause aerosol layer pathway results in insignificant enhancements of AOD

during JJAS by ~0.01- 0.02 over south Asia compared to the observed climatological mean

AOD (~0.6) [*Vernier et al.*, 2015; *Yu et al.*, 2015]. Thus, contributions of Asian tropopause

aerosol layer to the observed positive gradients (Figure 3) can be assumed to be negligible.

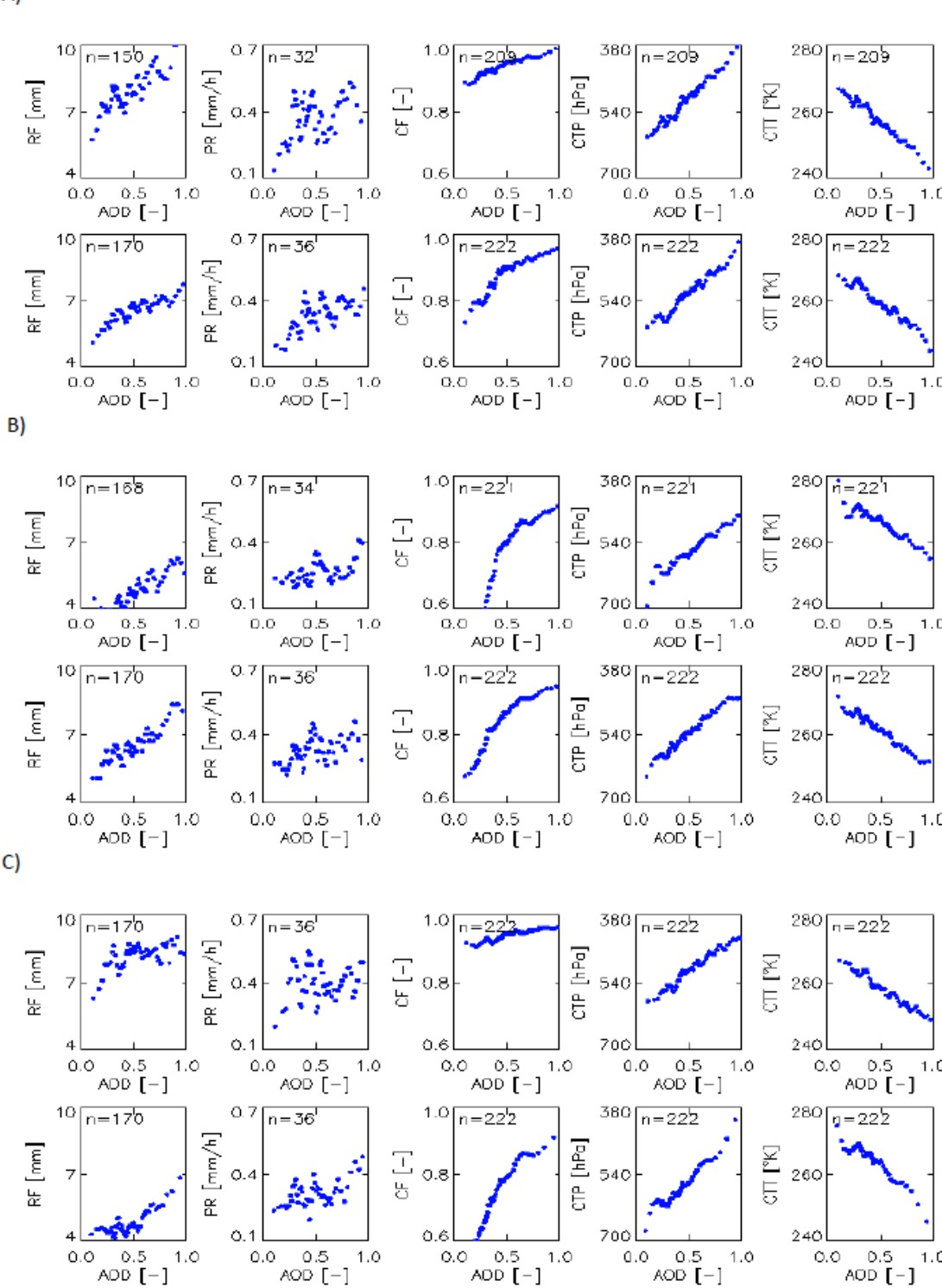

**Figure 11.** Associations of accumulated daily rainfall, precipitation rate, cloud fraction, cloud top pressure and cloud top temperature with AOD. (A) Data slicing by the wind shear for the lower regime (0-33%, Top) and the higher regime (67-100%, Bottom). (B) Same as A), except data slicing by the relative humidity (C) Same as A), except data slicing by the geo-potential height. The methodology of creation of the scatter points were similar to that used for Figure 3. Each scatter point is the average of these equal 'n' numbers of data points mentioned in each respective panels.

### 3.3.2 Examining the influence of cloud contamination effect

Here, we used radiosonde observations from eight stations in ISMR (Table 2) to illustrate humidification effect on satellite retrieved AOD. The total number of cloudy profiles varied from 270 (Ranchi) to 1065 (Kolkata). The mean and standard deviation in RH for these selected profiles were calculated (for each station data) in two layers of 1.5 km and 3 km, from surface. The bias in mean RH between shallower and deeper clouds for each station is also presented in Table 2. The range of variation in mean RH for each layer has been presented in Table 3. We found that with increase in mean RH, the natural variance in RH decreased for both the layers within ISMR. The mean and standard deviation of RH in 1.5 km (3 km) layer was found to be 84.3±13.2% (84.7±13.5%) under cloudy conditions within ISMR. At the same time, the bias in mean RH (associated with vertical change in cloud layer height) in 1.5 km and 3.0 km layer was found to be 2.7 % and 2.5 %, respectively. It can be seen that the bias was negligible compared to the natural variation present in RH during cloudy conditions in ISMR. Using the parameterization developed in Bar-Or *et al*., (2012), the maximum change in AOD was estimated to be about 0.1 due to the humidification effect (Table 3). Thus, the uncertainties in our data analyzes due to aerosol humidification effect seems to be minimal. Note that the difference in clean and polluted conditions in this study (AOD of about 1.0) was nearly an order of magnitude higher than the estimated maximum change in AOD (~0.1) due to the humidification effect. Therefore, the observed positive associations between AOD and cloud/rainfall properties do not appear to be significantly affected by aerosol growth due to humidification during cloudy conditions. In fact, the observed negative relationship between AOD and increase in RH over ISMR (Figure 11) appears to dominate the otherwise expected higher hygroscopic growth of aerosols and supports the above argument.

**Table 2.** World Meteorological Organisation (WMO) index number of radiosonde stations (WMO#), station latitude (Lat.), longitude Lon.), elevation above mean sea level (Elev.), number of radisonde profiles (N), number of cloudy profiles ($N_{cloudy}$), Mean RH and bias in RH for 1.5 km layer ($RH_{1.5}$ and $RH_{1.5,bias}$, respectively.) and 3.0 km layer ($RH_{3.0}$ and $RH_{3.0,bias}$, respectively) and median of cloud layer height (CLH) for each of the 8 radiosonde stations used in humidification analysis. "±" indicates standard deviation.

| Station | Vizag | Kolkata | Bhubaneswar | Patna | Lucknow | Nagpur | Bhopal | Ranchi |
|---|---|---|---|---|---|---|---|---|
| WMO # | 43150 | 42809 | 42971 | 42492 | 42369 | 42867 | 42667 | 42701 |
| Lat. ($^o$N) | 17.43 | 22.39 | 20.15 | 25.36 | 26.45 | 21.06 | 23.17 | 23.19 |
| Lon. ($^o$E) | 83.14 | 88.27 | 85.50 | 85.06 | 80.53 | 79.03 | 77.21 | 85.19 |
| Elev. (m) | 3 | 6 | 46 | 60 | 128 | 310 | 523 | 652 |
| N (#) | 2007 | 2291 | 2306 | 1432 | 1751 | 1916 | 1725 | 1616 |
| $N_{cloudy}$ (#) | 770 | 1065 | 823 | 709 | 837 | 898 | 555 | 270 |
| $RH_{1.5}$ | 83±12 | 88±11 | 89±11 | 88±11 | 83±15 | 82±16 | 83±16 | 89±12 |
| $RH_{1.5,bias}$ | 1.96 | 0.95 | 1.5 | 0.6 | 6.1 | 3.8 | 6.4 | 1.4 |
| $RH_{3.0}$ | 82±12 | 86±14 | 88±12 | 87±13 | 84±15 | 84±15 | 84±16 | 87±14 |
| $RH_{3.0,bias}$ | 0.5 | 1.5 | 0.8 | 1.4 | 5.1 | 3.8 | 6.4 | 0.3 |
| CLH (m) | 13810 | 14539 | 14455 | 14413 | 13430 | 9658 | 6343 | 9219 |

**Table 3**: Estimating change in AOD [∆AOD ] due to variation in RH. The hygroscopicity parameter, *k* used in the estimation was taken as 0 .1 and 0.2 to illustrate minimum and maximum change due to change in aerosol properties.

| | Layer 1.5 km | Layer 3 km |
|---|---|---|
| Range of mean RH | 72 - 97 | 71 - 98 |
| RH scaled as distance from nearest cloud | 0.02-0.13 | 0.02-0.12 |
| Maximum ∆ (AOD) for *k* = 0.1 (0.2) | ~0.05 (0.1) | ~0.05 (0.1) |

### 3.3.3 Investigating the effect of wet scavenging on aerosol-rainfall associations

Contrary to the positive aerosol-cloud-rainfall associations shown by many satellite data studies across the globe, recent studies have illustrated a negative aerosol-rainfall association mainly over tropical ocean region based on reanalysis dataset and global model simulations. This difference in sign of the association in modelling studies is mainly attributed to inclusion of wet scavenging effect in models and probable lack of the same in satellite samples [*Grandey et al.*, 2013; *Grandey et al.*, 2014; *Yang et al.*, 2016]. However, global modelling studies have their own inherent limitations and uncertainties in addressing aerosol-cloud-rainfall associations. Due to computational constraints, the global model

simulations use grids with coarse spatial resolution (~ 200 km) and falls short of explicitly resolving the fine-scale cloud processes. Moreover, the convection parameterizations used to simulate cloud formation generally do not parameterize the aerosol indirect effect on clouds and thus on rainfall. On the contrary, the observed relations using satellite datasets are at fine

scale and inclusive of the aerosol indirect effect. As a representative analysis, collocated AOD-rainfall measurements at hourly temporal resolution over IITK was used to illustrate the association between aerosol-rainfall with and without wet scavenging effect. Positive association was found between rainfall amount and mean AOD values measured before the starting of rain events over IITK (NWS_IITK; red line in Figure 12). Similar association was

also found when all the available collocated AOD-rain amount samples over IITK were correlated (Cyan color line in Figure 12), but the gradient was reduced by almost 50 % compared to that of NWS_IITK. Thus, positive association between aerosol-rainfall was evident even with the inclusion of wet scavenging effect in the sampling. Grandey et al., 2013 [*Grandey et al.*, 2013] have also shown similar amount of contribution of wet scavenging

effect on the positive aerosol-cloud association. Correlation of MODIS-AOD with RF (black line in Figure 12) and PR (blue line in Figure 12) values over the IITK grid also illustrated positive association between aerosol and rainfall similar to the observed associations in Figure 3. High anthropogenic aerosol emission rate at surface [*Bond et al.*, 2004] and the rapid aerosol build-up within a few hours after the individual rainfall event over ISMR [*Jai*

*Devi et al.*, 2011] might contribute towards reducing the impact of wet scavenging effect on the aerosol-cloud-rainfall analysis over ISMR. This argument is also supported by a pattern seen in model results that negative aerosol-cloud-rainfall associations were usually prominent over ocean regions and positive aerosol-cloud-rainfall associations were found over continental conditions in global simulations [*Grandey et al.*, 2013; *Grandey et al.*, 2014;

*Gryspeerdt et al.*, 2015; *Yang et al.*, 2016]. Unlike continental conditions, lack of high

emission rates at the ocean surface might also contribute to the dominant effect of wet scavenging on aerosol-cloud-rainfall association. In addition, the cloudy pixels where rainfall actually occurs under continental conditions are usually a small fraction of the total area within a 1° x 1° box, and therefore, the reduction in mean AOD value of the 1° x 1° box due to wet scavenging might not be a dominant phenomena affecting the aerosol-cloud-rainfall gradients in Figure 3. IITK-AERONET data analysis offers confidence to the observed position association for aerosol-cloud-rainfall, and that was not misrepresentation due to possible uncertainties involved for wet scavenging effect in using satellite retrieved AOD values. It indeed also showed that a more accurate representation of wet scavenging effect is essential to reduce uncertainty about the magnitude of the positive aerosol-rainfall gradient observed over ISMR.

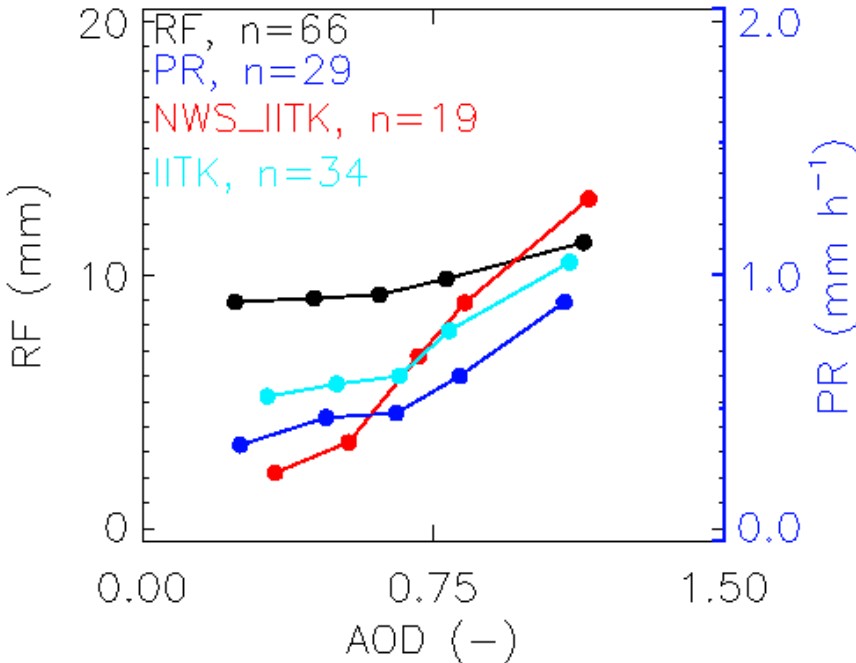

Figure 12: Associations of rainfall with collocated AERONET-AOD measurements (within ± 4 hours of the start/end of rainfall event) over IITK. The Cyan colorline illustrate the scenario with inclusion of wet scavenging effect (IITK) and the red color line illustrate the scenario with no wet scavenging effect (NWS_IITK). The association between daily rainfall and precipitation rate with MODIS-AOD over IITK grid is also shown  in black and blue color lines, respectively. In each case, all the rainfall-AOD samples were sorted as a function of corresponding AOD values into 5 bins of 20 percentiles each. Each scatter point is the average of each bin and have *n* number of data points.

## 4. Summary

In this study, long-term satellite and in-situ observational datasets were systematically analysed to get new insights in aerosol-cloud-rainfall associations over ISMR. An important finding is that the MODIS retrieved cloud properties (CF, CTP, CTT), IMD in-situ surface accumulated rainfall as well as TRMM retrieved precipitation rate illustrated a positive association with increasing aerosol loading. Additional selective analysis over smaller spatial region within ISMR and by separating the dataset into relatively shallower and deeper clouds also illustrated similar aerosol-cloud-rainfall associations, plausibly highlighting the robustness of these associations. A decrease in outgoing long wave radiation and increase in outgoing short wave radiation at the top of the atmosphere, with increase in aerosol loading further suggested deepening of cloud systems over ISMR.

Further, MODIS and CloudSat observed microphysical differences between low and high aerosol loading were investigated to gain process level understanding of the observed associations. Comparison of mean profiles of CTP-$R_e$ illustrated that increase in aerosol loading is associated with slower growth of $R_e$ with altitude, indicating reduction of coalescence efficiency and delay in initiation of warm rain. CloudSat retrieved profiles showed that the liquid water content increased under high aerosol loading, mainly the supercooled liquid droplets above the freezing level. Simultaneously, the observed mass concentration and effective radius of ice-phase hydrometeors increased manifold under high aerosol loading. We also performed three idealized supercell simulation of a typical heavy rainfall event over ISMR by varying initial CCN concentrations. Modelling results were found to be in-line with our observational findings, showing that CCN-induced initial suppression of warm phase processes along with increase in updraft velocity lead to movement of more water mass across freezing level resulting in enhancement of ice-phase

hydrometeor concentration and eventually in intensification of surface rainfall under high CCN loading.

We understand the limitation that influences of meteorological condition are ideally difficult to separate from that of aerosol on cloud-rainfall system. However, we have

systematically shown that the positive aerosol-cloud-rainfall associations were present even in narrow regimes of key cloud forming meteorological variables like RH, geopotential height and wind shear. Further, the ambiguity involved in humidification effect on retrieved AOD can also affect the positive gradients between aerosol and cloud-rainfall properties. Besides, AOD also suffers from substantial uncertainty in being representative of CCN

concentration near cloud base [*Andreae*, 2009]and in inclusion of wet scavenging effect in the AOD samples. These caveats may result in an overestimation of the observed positive gradients in aerosol-cloud-rainfall associations. Our analysis therefore cannot quantify the magnitude of gradients with confidence. However, this study certainly suggests a significant role of aerosol on rainfall properties via cloud invigoration over ISMR. As a future scope,

more observational studies at cloud formation and rain event time scales are warranted to accurately quantify the magnitude of aerosol-cloud-rainfall association over ISMR. Moreover, consideration of aerosol microphysical effects is essential for accurate prediction of monsoonal rainfall over this region of climatic importance.

**Acknowledgements**

This work was supported by India's Department of Science & Technology (DST) Climate Change Program (grant no. DST/CCP/(NET-1)-PR 22/2012). The rainfall gridded dataset is available with Indian Metrological Department and all satellite datasets (GDAS, CLOUDSAT, MODIS, CERES, TRMM) are available from respective online data archives.

Radiosondes were available online from university of Wyoming.

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
