# Peer review of "Investigation of aerosol-cloud-rainfall association over Indian Summer Monsoon region"

_Atmospheric Chemistry and Physics, 2016_

## Referee Comment (RC1) · Anonymous Referee #1 · 19 Dec 2016

The authors used 12 years of in-situ and satellite observations combined with model simulations to examine association of aerosol loading with cloud fraction, cloud top pressure, cloud top temperature, and daily surface rainfall over Indian summer monsoon region (ISMR). They found high aerosol loading might induce cloud invigoration and thereby increasing surface rainfall over the ISMR. This study contributes to address aerosol-cloud-rainfall associations over ISMR. Before this manuscript can be considered for publication, I have a few comments that need to be addressed by the authors.

My major concern of this paper is that the author found that impact of aerosols on cloud and rainfall dominated the aerosol-cloud-rainfall relationship over ISMR. However, many previous studies found that wet scavenging of aerosols by rainfall control

this relationship, while aerosol indirect effect could only perturb it (Grandey et al., 2013, 2014; Yang et al., 2016a,b). Why they are different from the results shown here?

Specific comments:

Page 1 Line 24: What is the seminal role? It needs to be expressed more specifically.

Page 3 Line 16: Aerosols interact with clouds and rainfall two ways. Aerosols impact clouds and precipitation, and clouds and precipitation can influence aerosols through wet scavenging processes (Quaas et al., 2010; Grandey et al., 2013, 2014; Gryspeerdt et al., 2015; Yang et al., 2016a,b,c). In introduction section, the authors only discussed aerosol impacts on cloud. They might add the cloud and rainfall influence on aerosols here.

Page 4 Line 11: Also, please also add description about impact of monsoon on aerosols (e.g. Corrigan et al., 2006; Lou et al., 2016).

Page 6 Line 8: This sentence is hard to follow. Please make it clear.

Page 6 Line 15: Could you also calculate the ratio of AODs>1.0 as same as 7% shown below?

Page 8 Line 20 and Page 9 Line 5: Why high and low AOD categories are different here?

Page 12 Line 10: aerosol humidification effect is not the only effect that causing co-variation of AOD- and rainfall. Engström and Ekman (2010) and Yang et al. (2016a) found wind speed also lead to co-variation of AOD-cloud and rainfall, it impact could be larger than aerosol humidification effect.

Page 12 Line 10: Also, I suggest adding impact of cloud-rainfall on AOD here as (3) and analysising it in result section.

Page 14 Line 17: Figure 3A. Is this figure the JJAS daily data for 2002–2013 (total 1464 samples)? How did you treat intra-seasonal variability of these variables? Please

[Figure]

make it clear here or in figure caption.

Figure 4: Again, please make clear how data used. What are these radiative fluxes, net, upward or downward?

Figure 7: Why not show Ex2?

Page 26 Line 21: The authors did not consider the impact of combination of meteorological fields. They may dampen each other. And the correlation in Figure 10 between different meteorological fields did not take into account the correlation among the meteorological variables.

Page 31 Line 2: Could you describe the method used in Bar-Or et al. (2012)?

Page 31 Line 16: This conclusion needs more cautiousness. The analysis using all-sky and clear-sky of CERES radiative fluxes may not represent aerosol direct and indirect effects. At lest, add region here.

Technical corrections:

Table 1: change 2002–13 to 2002–2013

Page 7 Line 8: I suggest changing abbreviation of rainfall. DRF is often used to represent direct radiative forcing.

Figure 2: What are these colored lines? Please add more information in in figure caption.

There are too many abbreviations in this study.

References:

Grandey, B. S., P. Stier, and T. M. Wagner (2013), Investigating relationships between aerosol optical depth and cloud fraction using satellite, aerosol reanalysis and general circulation model data, Atmos. Chem. Phys., 13(6), 3177–3184, doi:10.5194/acp-13-3177-2013.

Grandey, B. S., A. Gururaj, P. Stier, and T. M. Wagner (2014), Rainfall-aerosol relationships explained by wet scavenging and humidity, Geophys. Res. Lett., 41, 5678–5684, doi:10.1002/2014GL060958.

Yang, Y., L. M. Russell, S. Lou, Y. Liu, B. Singh, and S. J. Ghan (2016a), Rain-aerosol relationships influenced by wind speed, Geophys. Res. Lett., 43, doi:10.1002/2016GL067770.

Yang, Y., et al. (2016b), Impacts of ENSO events on cloud radiative effects in preindustrial conditions: Changes in cloud fraction and their dependence on interactive aerosol emissions and concentrations, J. Geophys. Res. Atmos., 121, doi:10.1002/2015JD024503.

Quaas, J., B. Stevens, P. Stier, and U. Lohmann (2010), Interpreting the cloud cover aerosol optical depth relationship found in satellite data using a general circulation model, Atmos. Chem. Phys., 10(13), 6129–6135, doi:10.5194/acp-10-6129-2010.

Yang, Y., Russell, L. M., Lou, S., Lamjiri, M. A., Liu, Y., Singh, B., & Ghan, S. J. (2016c). Changes in Sea Salt Emissions Enhance ENSO Variability. Journal of Climate, 29(23), 8575-8588.

Corrigan, C. E., V. Ramanathan, and J. J. Schauer (2006), Impact of monsoon transitions on the physical and optical properties of aerosols, J. Geophys. Res., 111, D18208, doi:10.1029/2005JD006370.

Lou, S., L. M. Russell, Y. Yang, L. Xu, M. A. Lamjiri, M. J. DeFlorio, A. J. Miller, S. J. Ghan, Y. Liu, and B. Singh (2016), Impacts of the East Asian Monsoon on springtime dust concentrations over China, J. Geophys. Res. Atmos., 121, 8137–8152, doi:10.1002/2016JD024758.

Engström, A., and A. M. L. Ekman (2010), Impact of meteorological factors on the correlation between aerosol optical depth and cloud fraction, Geophys. Res. Lett., 37, L11814, doi:10.1029/2010GL044361.

---

## Referee Comment (RC2) · Anonymous Referee #2 · 24 Dec 2016

The authors presented aerosol-cloud-rainfall associations over Indian derived using multiple datasets from satellite and meteorological stations. The subject of the paper is appropriate for ACP. However, the paper contains some issues indicated in the following comments. I recommend the acceptance of the paper for publication only after these issues being addressed by the authors.

1. Though the number of total samples is given in Figure 3, the number of samples in each AOD bin is not found in the figure and the text. The vast majority of cases may be with middle AOD, but the very clean cases (e.g. 0.01~0.1) and very heavy polluted cases may be relatively rare over the study region in Indian. Therefore, the representation of samples in both very clean and very polluted cases is unclear. Conclusions

based on a few extreme data points or from largely scattered data need further scrutiny. Such concerns are also in Figure 11.

2. The analysis based on the simulation work is irrelevant to the title of this paper "A long-term observational analysis of aerosol-cloud-rainfall associations over Indian Summer Monsoon region". Some modifications are necessary.
* * *

---

## Referee Comment (RC3) · Dr. M G (Referee) · 3 Jan 2017

Reviewer Comments on the paper entitled "**A long-term observational analysis of aerosol-cloud-rainfall associations over Indian Summer Monsoon region**" by Chandan Sarangi et al., 2016.

This paper examines the relationship that exists between aerosols and precipitation through aerosol-cloud-interaction. In general the paper is written well, analysed with all the available techniques for separation from meteorological effects, and necessary references have been cited properly, but it can be improved further by correcting the comments and concerns. Though the authors have taken efforts in doing such a laborious analysis, the anticipated results are more qualitative in nature due to the complexity with decoupling the meteorology. Hence, I suggest the reviewers to take up a major revision of the same by considering the comments given below. The detailed review comments are provided below.

**Major Comments**

1.  Page 4, Line 4: For a comprehensive review, discuss briefly on other studies which report evidences for increase in rainfall as a result of enhanced warming over IGP region due to aerosol radiative effects and associated dynamical feedbacks (Lau et al, 2006; Manoj et al., 2011 etc.).

2.  Page 9, line 5: Why $25^{th}$ and $75^{th}$ percentile between high and low AOD conditions were used? In section 2.3, the authors had adopted 33 and 67 percentiles. Why are these cut offs different, and what is the condition which these limits have been based up on?

3.  Page 11, Line 3: Since the authors have all the relevant meteorological profiles at hand from CAIPEEX experiment, why the assumption regarding exponentially decreasing temperature pulse of 3° C was used?

4.  Page 13, Lines 20-23: 'The height above the LCL where the theoretical temperature of a buoyantly rising moist parcel (following wet adiabatic lapse rate) becomes equal to the temperature of the environment is referred to as equilibrium level'. This statement is not exactly correct. The Level of Free Convection (LFC) also satisfies the above criteria. Hence it is correct to change as: 'The height above the **LFC**…. to as equilibrium level'.

5.  Page 17, Lines 5-6: (**a**) 'Thus, the aerosol indirect effect could be twice as high as aerosol direct effect over ISMR'. How did the authors estimate the indirect radiative effect? The Reviewer is doubtful about this statement here. Cloud formation is not simply as a result of aerosol indirect effect alone; however, it requires conducive thermodynamical and dynamical atmospheric processes too. Hence, the reported cooling by 30 $Wm^{-2}$ cannot be attributed to aerosol indirect effect alone, if the authors estimate the indirect effect by simply sorting AODs under cloudy conditions, and estimate the indirect forcing. (**b**) Reported cut-off values of AOD $\approx$ 0.3 illustrates that up to 0.3 AODs, the indirect effect dominates and beyond this limit the aerosol-radiation interaction effect dominates. No mention about this cut off is mentioned in this paper.

6.  Figure 6. Each line colour in Figure Caption given for Ex1 is wrong compared with those given in the Figure itself. Please correct. Same for other figures too (e.g. Figure 8)

7.  Page 21, Line 5: Is the vertical updraft velocity only 0.2 cm/s, when the convection is strong? Or is it in the unit of meter/second (instead of cm/s)?

8.  Page 26, Lines 15-19: Give a discussion on whether aerosol invigoration leads to increase in total rainfall averaged over all grids and time, or if it leads to a redistribution of rain with suppressed rain at initial time, and enhanced precipitation at a later stage so that total surface precipitation is nearly conserved.

9.  Figure 10 needs precise description for a general reader to comprehend the basic idea, especially about the x-axis, and the shaded region.

10. Page 26, Lines 22-25: A major drawback of the correlation analysis here is that it represents simultaneous correlation. However, aerosol build up might have taken place prior to cloud maturity and rain initiation, and subsequently could have reduced due to cloud scavenging and wet removal. A lag correlation analysis at cloud formation time scales could have been more meaningful here.

11. Section 3.4 could be merged with an earlier description of AOD retrieval errors associated with contamination due to RH. This section is a repetition.

**Minor Comments**

1.  Title: 'Association' instead of 'Associations'.
2.  Abstract, Line 31: Change to 'Simulated microphysics also illustrated *that* the…'
3.  Abstract, Line 36: Correct as: 'While the meteorological variability influence*s*'
4.  Abstract, Line 37: Change to 'association' instead of 'associations'.
5.  Page 2, Line 9: Remove comma (,) after 'cloud base'.
6.  Page 3, Lines 4 & 15; Page 9, Line 20, and many places: Correct 'AP Khain et al.' to 'Khain et al.'.
7.  Page 4, Line 2: Replace 'as well as' by 'and'.
8.  Reference required: '…lower available spatial resolution (i.e. $0.25\degree \times 0.25\degree$) was in general biased to smaller clouds..'.
9.  Page 9, line 1: Correct: 'CLOUD-aerosol Lidar and infrared pathfinder SATellite (not *CloudSat*, but *CALIPSO*)'.
10. Page 22, line 18: Correct: 'droplet spectral'.
11. Page 30, Lines 5: Remove 'other'.
12. Page 33, Lines 5: Change to 'found *to be* in-line…'

---

## Author Comment (AC1) · 18 Feb 2017

**Author's Response to Reviewer #2**

The authors presented aerosol-cloud-rainfall associations over Indian derived using multiple datasets from satellite and meteorological stations. The subject of the paper is appropriate for ACP. However, the paper contains some issues indicated in the following comments. I recommend the acceptance of the paper for publication only after these issues being addressed by the authors.

**Response**: We are thankful to the reviewer for the thorough reading of our manuscript. We have addressed all the comments and suggestions provided by the reviewer. Our point-to-point responses for the specific comments are mentioned below in blue color. The subsequent changes and additions made in the revised manuscript against each comment are shown in red color.

1. Though the number of total samples is given in Figure 3, the number of samples in each AOD bin is not found in the figure and the text. The vast majority of cases may be with middle AOD, but the very clean cases (e.g. 0.01~0.1) and very heavy polluted cases may be relatively rare over the study region in Indian. Therefore, the representation of samples in both very clean and very polluted cases is unclear. Conclusions based on a few extreme data points or from largely scattered data need further scrutiny. Such concerns are also in Figure 11.

**Response**: We have used equal number of samples in each AOD bin for the correlation analysis in Figure 3 and 11. Therefore, the representation of samples in both very clean and very polluted cases is equal. For better clarity and ease of understanding, we have mentioned the methodology used to create these plots explicitly in the caption of Figure 3, 4 and 11 in the revised manuscript.

Caption of **Figure 3.** Associations of (A) daily rainfall, (B) precipitation rate, (C) cloud fraction, (D) cloud top pressure, and (E) cloud top temperature with AOD. The collocated data points for these five variables (A-E) were sorted as a function of AOD over ISMR during JJAS 2002-2013. The total number of collocated data points are then used to create 50 bins with 'n' numbers of data points (2 percentile) each and averaged. These 50 scatter points are shown in the respective panels.

2. The analysis based on the simulation work is irrelevant to the title of this paper "A long-term observational analysis of aerosol-cloud-rainfall associations over Indian Summer Monsoon region". Some modifications are necessary.

**Response**: As suggested we have slightly revised the title of the paper to better convey the relevance. The revised title reads as -"Investigation of aerosol-cloud-rainfall association over Indian Summer Monsoon region "

In this study, we used 12 years of in-situ and satellite observations to examine association of aerosol loading with cloud fraction, cloud top pressure, cloud top temperature, and daily surface rainfall over Indian summer monsoon region (ISMR). We found high aerosol loading might induce cloud invigoration thereby increasing surface rainfall over the ISMR. The physical mechanisms of these relationships were better illustrated by performing numerical experiments on WRF-SBM platform using thermodynamic conditions typical of ISMR. Further, we have also examined the associations from different possible explanations.

---

## Author Comment (AC2) · 18 Feb 2017

**Author's Response to Reviewer #1**

The authors used 12 years of in-situ and satellite observations combined with model simulations to examine association of aerosol loading with cloud fraction, cloud top pressure, cloud top temperature, and daily surface rainfall over Indian summer monsoon region (ISMR). They found high aerosol loading might induce cloud invigoration and thereby increasing surface rainfall over the ISMR. This study contributes to address aerosol-cloud-rainfall associations over ISMR. Before this manuscript can be considered for publication, I have a few comments that need to be addressed by the authors.

**Response**: We are grateful to the reviewer for the thorough reading of our manuscript. We have addressed all the comments and suggestions provided by the reviewer. Our point-to-point response to each comment is indicated below in blue color. The subsequent changes and additions made in the revised manuscript in response to each comment are shown in red color.

My major concern of this paper is that the author found that impact of aerosols on cloud and rainfall dominated the aerosol-cloud-rainfall relationship over ISMR. However, many previous studies found that wet scavenging of aerosols by rainfall control this relationship, while aerosol indirect effect could only perturb it (Grandey et al., 2013, 2014; Yang et al., 2016a,b). Why they are different from the results shown here?

**Response**: We agree with the reviewer's concern about the difference in results between the two broad approaches (modeling analysis and data analysis using satellite retrievals) used for studying aerosol-cloud-rainfall associations. However, many studies (as mentioned in the Introduction section) have shown the aerosol induced changes to coupling between the microphysical and dynamical processes in the cloud and in the cloud field to explain the positive link while wet scavenging could not explain most of the effect. We have included a discussion in this context in the revised manuscript about the possible causes of these differences as well as about the limitations and uncertainties involved in both approaches regarding the competing effects of wet scavenging and aerosol microphysical modifications. Additionally, we have also analyzed aerosol-rainfall association with and without wet scavenging effect over Kanpur using AERONET-measured AOD and rain gauge measured rainfall at an hourly resolution. This analysis illustrated that the positive aerosol-rainfall association exists despite exclusively including wet scavenging effect thereby further strengthening our argument. Nevertheless, in agreement with the reviewer's statement, it has reduced the gradient of the association.

The added discussion and results in manuscript in response to the reviewer's comment for "Page 12 Line 10" are mentioned below.

**Specific comments:**

Page 1 Line 24: What is the seminal role? It needs to be expressed more specifically.

**Response**: We have included the following statement in the revised manuscript (Page 1 Line 22):

A decrease in outgoing longwave radiation and increase in reflected shortwave radiation at the top of the atmosphere with an increase in aerosol loading further indicates a possible seminal role of aerosols in deepening of cloud systems.

Page 3 Line 16: Aerosols interact with clouds and rainfall two ways. Aerosols impact clouds and precipitation, and clouds and precipitation can influence aerosols through wet scavenging processes (Quaas et al., 2010; Grandey et al., 2013, 2014; Gryspeerdt et al., 2015; Yang et al., 2016a,b,c). In introduction section, the authors only discussed aerosol impacts on cloud. They might add the cloud and rainfall influence on aerosols here.

**Response**: We have included the following discussions about wet scavenging effect on aerosols in the revised manuscript (Page 3, Line 16):

Moreover, clouds and precipitation can also interact with aerosols through wet scavenging process [*Grandey et al.*, 2013; *Grandey et al.*, 2014;*Yang et al.*, 2016]. Global model simulations illustrated that wet scavenging can cause a strong negative cloud fraction-AOD correlation over the tropics [*Grandey et al.*, 2013]. Wet scavenging effect can also generate similar negative rain rate-AOD association in the tropical and mid-latitude oceans [*Grandey et al.*, 2014].

Page 4 Line 11: Also, please also add description about impact of monsoon on aerosols (e.g. Corrigan et al., 2006; Lou et al., 2016).

**Response**: We have included the following discussions in the revised manuscript (Page 5, Line 1):

Conversely, summer monsoon plays an important role in determining variation in aerosol loading over India by bringing clean marine air and wet scavenging, which are as important as emission in determining aerosol concentration [*Li et al.*, 2016]. It has also been shown that aerosols over the Indian Ocean interplay with seasonal changes over ISMR[*Corrigan et al.*, 2006].

Page 6 Line 8: This sentence is hard to follow. Please make it clear.

**Response**: We have revised the following sentence in the revised manuscript for clarity (Page 7, Line 1).

RF as well as PR datasets were linearly re-gridded to the $1^{o}\times1^{o}$ grid for consistency in our correlation analysis.

Page 6 Line 15: Could you also calculate the ratio of AODs>1.0 as same as 7% shown below?

**Response**: The percentage of AOD values > 1 was ~5%. We have included this information in the revised manuscript Page 6 Line 21.

Page 8 Line 20 and Page 9 Line 5: Why high and low AOD categories are different here?

**Response**: The criteria for low and high AOD categories are same in the entire study as is mentioned in Page 8 Line 20. The concerned sentence at line 5 Page 9 does not state the criteria for segregating CLOUDSAT profiles into low and high aerosol bins. It is meant to describe the variability of the microphysical variables (thin lines representing the 25[th] and 75[th] percentile within each of the two AOD bins in Figure 5B). We have modified the sentence in the revised manuscript (Page 9 Line 19) as below for clarity.

The mean microphysical variables along with their variability (profiles indicating 25[th] and 75[th] percentile) for low and high aerosol bins were plotted against altitude to visualize the net increase or decrease in liquid-phase water content, ice-phase water content and size of ice-phase hydrometeors at different altitudes with increase in aerosol loading.

Page 12 Line 10: aerosol humidification effect is not the only effect that causing covariation of AOD- and rainfall. Engström and Ekman (2010) and Yang et al. (2016a) found wind speed also lead to co-variation of AOD-cloud and rainfall, it impact could be larger than aerosol humidification effect.

**Response**: The correlation of AOD, RF and CF with wind speed at different heights is shown in Figure 10. The analysis over ISMR illustrates that wind speed in the lower troposphere is not a dominant factor. Moreover, positive correlation between AOD-wind speed is associated with a negative correlation between CF/RF- wind speed at the same altitude. This is different from the findings of Engström and Ekman (2010) and Yang et al. (2016a), where, increase in wind speed at 10m height is strongly correlated with increase in both AOD and accumulated rainfall/precipitation rate. A possible reason for this may be that unlike our study, both, Engström and Ekman (2010) and Yang et al. (2016a) were performed exclusively over oceans. Wind speed can increase the aerosol loading over the ocean (more production of sea-spray aerosol) but over polluted land regions with local sources it could dilute the aerosol concentration.

We found a strong impact of wind shear on AOD-cloud-rainfall association, consistent with aerosol-induced cloud invigoration as shown by previous studies (Fan et al., 2009). So we have considered wind shear as a key meteorological factor over ISMR and have analysed its effect on observed aerosol-cloud-rainfall analysis.

[Figure]

**Figure 10**.Correlation coefficients of accumulated daily rainfall, AOD and cloud fraction with six GDAS meteorological variables over ISMR. Different color shades along the x-axis indicate corresponding meteorological variable and each color shade has 21 divisions which represents corresponding 21 model pressure levels from 100 hPa to1000 hPa (left to right). Correlation analysis was performed at each model pressure level with all collocated samples between two variables (e.g. temperature and RF for x-axis values of 1-21 in case of red color line) over ISMR region for JJAS 2002-2013.

Page 12 Line 10: Also, I suggest adding impact of cloud-rainfall on AOD here as (3) and analysising it in result section.

**Response**: As per your suggestion, we have added analysis and discussion on the wet scavenging impact of cloud-rainfall on AOD. Corresponding modifications in each section are mentioned below in red color.

Section 2.5

(3) Underestimation of wet scavenging effect on satellite retrieved AOD values [Grandey et al., 2013;2014].

Section 2.5.3

[revised manuscript text omitted]

Page 14 Line 17: Figure 3A. Is this figure the JJAS daily data for 2002–2013 (total 1464 samples)? How did you treat intra-seasonal variability of these variables? Please make it clear here or in figure caption.

**Response**: We have modified the caption of Figure 3 in the revised manuscript to provide information about the sampling used in these correlations. The modified caption is shown below

Caption of **Figure 3.**Associations of (A) daily rainfall, (B) precipitation rate, (C) cloud fraction, (D) cloud top pressure, and (E) cloud top temperature with AOD. The collocated data points for these five variables (A-E) were sorted as a function of AOD over ISMR during JJAS 2002-2013. The total number of collocated data points are then used to create 50 bins with 'n' numbers of data points (2 percentile) each and averaged. These 50 scatter points are shown in the respective panels.

Moreover, we have also created figures similar to Figure 3, but separately for June, July, August and September. The comparison of these plots illustrate that the positive aerosol-cloud-rainfall association exists in each month separately, however, the range and variability in the association varies. As our focus is to illustrate the qualitative association, we have merged the data from all

the four monsoon months into one plot to examine the seasonal pattern. This also reduces the spread of the regression by increasing the number of samples (n) per AOD bin. We have stated this information in the modified manuscript as below (in red) at **Page 18 Line 7**.

Analysis of individual months viz; June, July, August and September also illustrated similar positive associations as seen in Figure 3 indicating negligible intra-seasonality in the observed associations.

[Figure]

**Figure:** Associations of daily rainfall (A), precipitation rate (B), cloud fraction (C), cloud top pressure (D) and cloud top temperature (E) with AOD. The daily collocated measurements of these five variables with AOD from all the grids within ISMR during June (Row1; top row), July (Row2), August (Row3), and September (Row4; bottom row) monthsof 2002-2013 are used. They are sorted as a function of collocated AOD values (each variable separately).The total

number of collocated data points are then used to create 50 bins with 'n' numbers of data points (2 percentile) each and averaged. These 50 scatter points are shown in the respective panels.

Figure 4: Again, please make clear how data used. What are these radiative fluxes, net, upward or downward?

**Response**: We have modified the caption of Figure 4 as shown below to incorporate the suggested information.

**Figure 4**. Association of CERES retrieved incoming shortwave (SW) and outgoing longwave (LW) radiation with AOD for (A) all-sky and (B) clear-sky scenario over ISMR during JJAS 2002-2013.  The collocated data points for both SW and LW as a function of AOD were first sorted. The total number of collocated data points are then used to create 50 bins with 'n' numbers of data points (2 percentile) each and averaged. These 50 scatter points are shown in the respective panels.

Figure 7: Why not show Ex2?

**Response**: We have included Ex2 in Figure 7. We had not included Ex2 previously to reduce the density of the figures as the results are proportional between Ex1, Ex2 and Ex3.

[Figure]

Figure 7: A) Mean droplet  $R_e$ versus CTP for low (Ex1; blue), medium (Ex2; black) and high (Ex3; red) CCN scenario. B) Droplet size distribution spectra of Ex1 (blue), Ex2 (black) and Ex3 (red) simulations at 700 hPa (dashed lines) and 300 hPa (solid lines). The corresponding effective radius values are mentioned in the legends in square brackets. Fractional contribution is calculated by dividing the mass concentration of each bin with the total mass concentration.

Page 26 Line 21: The authors did not consider the impact of combination of meteorological fields. They may dampen each other. And the correlation in Figure 10 between different meteorological fields did not take into account the correlation among the meteorological variables.

**Response**: We agree with the reviewer that many of the meteorological variables are not orthogonal to all others and therefore can have significant correlation between them. Nevertheless, when classifying the atmosphere in accord to its cloud formation potential, most of the variables indicate the same trend, i.e. for deeper clouds we will find higher CAPE levels, low

geopotential height, stronger updrafts, higher relative humidity in the cloud layers, stronger moist conversion, etc. Conflicting trends are possible but based on our analysis the trend is quite uniform in all key variables.

As per suggestion of the reviewer, we have also analysed the impact of combining the key meteorological factors in our analysis. The results are shown in Figure A below. The entire dataset was divided into meteorological slices to represent 8 regimes as shown in the table below. The positive association between aerosol-cloud-rainfall existed in each sub regime, however, the relationship indeed dampened due to the orthogonal effect of ambient meteorological conditions as also discussed in the manuscript. We have included information about this analysis in the revised manuscript at **Page 31 Line 16**

Page 31 Line 16

We have also considered the combined effect of all the three key meteorological variables by dividing the datasets into 8 regimes (alternate combination of higher and lower bins of RH,WS and GPH). Our analysis illustrated (Figure not shown) similar results as seen in Figure 11; positive aerosol-cloud-rainfall association was evident in all the 8 sub-regimes along with distinct orthogonal effect of ambient meteorological conditions.

Table: Segregation criteria of the 8 regimes created for investigating the orthogonal effect of meteorology on aerosol-cloud-rainfall analysis combining all the three key factors. The association in each regime is shown in figure B below.

| Regimes | RH | Wind shear | GPH |
|---------|-----|-----------|-----|
| R1 | High (>67 percentile) | High (>67 percentile) | High (>67 percentile) |
| R2 | High (>67 percentile) | High (>67 percentile) | Low (<33 percentile) |
| R3 | High (>67 percentile) | Low (<33 percentile) | High (>67 percentile) |
| R4 | High (>67 percentile) | Low (<33 percentile) | Low (<33 percentile) |
| R5 | Low (<33 percentile) | High (>67 percentile) | High (>67 percentile) |
| R6 | Low (<33 percentile) | High (>67 percentile) | Low (<33 percentile) |
| R7 | Low (<33 percentile) | Low (<33 percentile) | High (>67 percentile) |
| R8 | Low (<33 percentile) | Low (<33 percentile) | Low (<33 percentile) |

[Figure]

[Figure]

Figure A: Associations of accumulated daily rainfall, precipitation rate, cloud fraction, cloud top pressure and cloud top temperature with AOD for meteorology regime **A) R1, B) R2, C) R3, D) R4, E) R5, F) R6, G) R7, H) R8.** The daily collocated measurements of these variables (except PR) with AOD from all the grids within ISMR during JJAS, 2002-2013 are sorted as a function of collocated AOD values (each variable separately), divided into 25 bins of 4 percentile each and averaged. In case of PR-AOD association, the number of sample points were greatly reduced within each regime so the dataset was divided into 5 equal bins of 20 percentile each and samples averaged. The number of samples (n) corresponding to each scatter point are indicated on the plot. The total number of samples used in these separate associations are equal to n times 50.

Page 31 Line 2: Could you describe the method used in Bar-Or et al. (2012)?

**Response**: We have described the methodology of Bar-Or et al. (2012) in detail in Methodology section Page 15 Line 5.

Page 31 Line 16: This conclusion needs more cautiousness. The analysis using all-sky and clear-sky of CERES radiative fluxes may not represent aerosol direct and indirect effects. At least, add region here.

**Response**: We have removed the statement in the revised manuscript.

**Technical corrections:**

Table 1: change 2002–13 to 2002–2013

Response: We have modified the caption abbreviation as suggested.

Page 7 Line 8: I suggest changing abbreviation of rainfall. DRF is often used to represent direct radiative forcing.

Response: We have replaced DRF with RF in the manuscript and figures.

Figure 2: What are these colored lines? Please add more information in figure caption.

Response: We have modified the caption of Figure 2 with detailed information.

[Figure]

**Figure 2**.Skew-T - log-P diagram illustrating the initial conditions of dew point temperature (red hashed line) and atmospheric temperature (red solid line) used in all the three WRF-SBM idealized simulations. Blue, yellow, green, black and purple lines indicate lines of constant temperature (isotherm), potential temperature and equivalent potential temperature, pressure (isobar), and saturation mixing ratio, respectively.

There are too many abbreviations in this study.

We have tried to keep them to minimum.

---

## Author Comment (AC3) · 18 Feb 2017

**Authors' Response to Reviewer #3**

This paper examines the relationship that exists between aerosols and precipitation through aerosolcloud-interaction. In general the paper is written well, analysed with all the available techniques for separation from meteorological effects, and necessary references have been cited properly, but it can be improved further by correcting the comments and concerns. Though the authors have taken efforts in doing such a laborious analysis, the anticipated results are more qualitative in nature due to the complexity with decoupling the meteorology. Hence, I suggest the reviewers to take up a major revision of the same by considering the comments given below.

**Response**: We are very grateful to the Dr Manoj M. G. for his thorough reading and useful suggestions for improvement of our manuscript. We have addressed all the comments and suggestions provided by the reviewer. Our point-to-point responses for the specific comments are mentioned below in blue color. The subsequent changes and additions in the revised manuscript against each comment are shown in red color.

The detailed review comments are provided below.

Major Comments

1. Page 4, Line 4: For a comprehensive review, discuss briefly on other studies which report evidences for increase in rainfall as a result of enhanced warming over IGP region due to aerosol radiative effects and associated dynamical feedbacks (Lau et al, 2006; Manoj et al., 2011 etc.).

**Response**: We have added a detailed discussion on aerosol radiative impact on Indian monsoon rainfall in Introduction (Page 4 Line 6) as mentioned below.

Recent studies based on aerosol direct effect have shown different plausible pathways of aerosol impact on rainfall. Lau and Kim (2006) [*Lau and Kim*, 2006] have shown that aerosol-induced atmospheric heating over Himalayan slopes and Tibetan plateau during monsoon onset period, intensifies the northward shift of Indian summer monsoon, causing reduction in rainfall over ISMR. On the other hand, high aerosol loading also induces a solar dimming (absorbing) effect at surface [*Ramanathan and Carmichael*, 2008; *Ramanathan et al.*, 2001], which can alter the land-ocean thermal gradient and weaken the meridional circulation, resulting in a drying trend in seasonal rainfall during Indian summer monsoon [*Bollasina et al.*, 2011; *Ganguly et al.*, 2012]. Presence of higher concentrations of absorbing aerosols over North India is shown to induce a stronger north–south temperature difference which fosters enhancement in moisture convergence from ocean and a transition from a break spell of ISM to an active spell of ISM [*Manoj et al.*, 2011]. Further, this aerosol radiative effect causes increase in the moist static energy, invigoration of convection and eventually more rainfall over India during the following active phase [*Hazra et al.*, 2013; *Manoj et al.*, 2011].

2. Page 9, line 5: Why 25th and 75th percentile between high and low AOD conditions were used? In section 2.3, the authors had adopted 33 and 67 percentiles. Why are these cut offs different, and what is the condition which these limits have been based up on?

**Response**: The criteria for low and high AOD categories are same in the entire study as is mentioned in Page 8 Line 20. The concerned sentence at line 5 Page 9 does not state the criteria of segregating CLOUDSAT profiles into low and high aerosol bins. It is meant to describe the

variability of the microphysical variables (e.g. thin lines representing the $25^{th}$ and $75^{th}$ percentile within each of the two AOD bins in Figure 5B). We have modified the sentence in the revised manuscript (Page 9 Line 19) as below for clarity.

The mean microphysical variables along with their variability (profiles indicating $25^{th}$ and $75^{th}$ percentile) for low and high aerosol bins were plotted against altitude to visualize the net increase or decrease in liquid-phase water content, ice-phase water content and size of ice-phase hydrometeors at different altitudes with increase in aerosol loading.

3. Page 11, Line 3: Since the authors have all the relevant meteorological profiles at hand from CAIPEEX experiment, why the assumption regarding exponentially decreasing temperature pulse of 3° C was used?

**Response**: CAIPEEX experiment was localized over Bareilly during 23rd August 2009, but we intend to simulate a storm over Patna, so we have not used CAIPEEX meteorological conditions in our simulation. We have used CCN spectra from CAIPEEX measurements only as a guideline for probable CCN concentrations over the region. For the SBM simulation Radiosonde-obtained initial thermodynamic conditions from Patna IMD station were used as initial conditions to simulate the environment and microphysical variability. An exponential $3^0$C temperature pulse is employed intrinsically in the SBM model instantaneously to trigger the initiation of parcel rise from surface as mentioned in the basic papers of the model. We have included references for this in the revised manuscript.

4. Page 13, Lines 20-23: 'The height above the LCL where the theoretical temperature of a buoyantly rising moist parcel (following wet adiabatic lapse rate) becomes equal to the temperature of the environment is referred to as equilibrium level'. This statement is not exactly correct. The Level of Free Convection (LFC) also satisfies the above criteria. Hence it is correct to change as: 'The height above the LFC…. to as equilibrium level'.

**Response**: We have replaced LCL with LFC as suggested.

5. Page 17, Lines 5-6: (a) 'Thus, the aerosol indirect effect could be twice as high as aerosol direct effect over ISMR'. How did the authors estimate the indirect radiative effect? The Reviewer is doubtful about this statement here. Cloud formation is not simply as a result of aerosol indirect effect alone; however, it requires conducive thermodynamical and dynamical atmospheric processes too. Hence, the reported cooling by 30 Wm-2 cannot be attributed to aerosol indirect effect alone, if the authors estimate the indirect effect by simply sorting AODs under cloudy conditions, and estimate the indirect forcing. (b) Reported cutoff values of AOD ≈ 0.3 illustrates that up to 0.3 AODs, the indirect effect dominates and beyond this limit the aerosol-radiation interaction effect dominates. No mention about this cut off is mentioned in this paper.

**Response**:

(a) We have removed this interpretation in the revised manuscript. The main focus of the AOD-CERES analysis is to illustrate the deepening of cloud systems with depth. So we have modified the sentence in **Page 19 Line 22** accordingly. We have also removed similar interpretation from the conclusions.

Quantitatively, the net cooling per unit increase in AOD (Figure 4B) under clear sky scenario was ~13 W/m$^2$,whereas the net cooling for same change in AOD under cloudy condition was twice more than that under clear sky scenario i.e. ~30 W/m$^2$.

(b) A discussion in this context is present in the manuscript on Page 17 Line 17.

Aerosol-cloud studies have reported reduction in cloudiness under high AOD for regions with high absorbing aerosol loading [*Koren et al.*, 2004; *Small et al.*, 2011]. Widespread cloud coverage over ISMR (CF of ~0.75 for AOD ~0.3 in Figure 3) induces substantial reduction in the incoming solar radiation [*Padma Kumari and Goswami*, 2010], which may result in reduced interaction between absorbing aerosols and shortwave radiation. This explains that, despite the high emission rate of absorbing aerosols over ISMR [*Bond et al.*, 2004], the aerosol-induced cloud inhibition effect seemed to have been reduced to a second order process during Indian summer monsoon.

6. Figure 6. Each line colour in Figure Caption given for Ex1 is wrong compared with those given in the Figure itself. Please correct. Same for other figures too (e.g. Figure 8)

**Response**: These mistakes are corrected in the revised manuscript.

7. Page 21, Line 5: Is the vertical updraft velocity only 0.2 cm/s, when the convection is strong? Or is it in the unit of meter/second (instead of cm/s)?

**Response**: The vertical velocity is indeed in m/s. We have corrected this in the revised manuscript.

8. Page 26, Lines 15-19: Give a discussion on whether aerosol invigoration leads to increase in total rainfall averaged over all grids and time, or if it leads to a redistribution of rain with suppressed rain at initial time, and enhanced precipitation at a later stage so that total surface precipitation is nearly conserved.

**Response**: Aerosol-induced cloud invigoration leads to increase in accumulated rainfall throughout the storm domain as seen in the Figure below. The initial suppression of warm rain favors transport of more water mass to higher altitudes and the formation of bigger and deeper clouds, which eventually result in enhanced rainfall. In this context, we have added a description as below at **Page 24  Line 22.**

However, the increase in rainfall amount with increase in CCN concentration in later stage of simulation was manifold compared to the initial suppression of warm rainfall eventually leading to the enhancement of accumulated rainfall throughout the storm domain(Figure not shown).

[Figure]

Figure: Accumulated rainfall after t=90 min (top row) , t=120 min (middle row) and t=150 min (bottom row) for Ex1 (left col.), Ex2 (middle col.) and Ex3 (right col.) simulations

9. Figure 10 needs precise description for a general reader to comprehend the basic idea, especially about the x-axis, and the shaded region.

**Response**: We have revised the caption of figure 10 as mentioned below to improve clarity.

Figure 10: Correlation coefficients of accumulated daily rainfall, AOD and cloud fraction with various GDAS meteorological variables over ISMR. Different color shades along the x-axis illustrate different meteorological variables and each color shade has 21 divisions which represent corresponding 21 model pressure levels from 100 hPa to 1000 hPa. Correlation analysis was performed at each model pressure level with all collocated samples (of the two variables used in the analysis) over ISMR region for JJAS, 2002-2013.

10. Page 26, Lines 22-25: A major drawback of the correlation analysis here is that it represents simultaneous correlation. However, aerosol build up might have taken place prior to cloud

maturity and rain initiation, and subsequently could have reduced due to cloud scavenging and wet removal. A lag correlation analysis at cloud formation time scales could have been more meaningful here.

**Response**: The reviewer is correct in stating that a lag correlation analysis at cloud formation/rainfall time scale will be interesting and insightful about the effect of wet scavenging and subsequent aerosol build-up effect. In the revised manuscript we have used long-term aerosol-rainfall ground based measurements over Kanpur to redo the analysis at hourly time scale and check the impact of wet scavenging. We have also added this analysis to results and discussion in our revised manuscript as suggested by Reviewer 1. The results and additions are mentioned below. We found that positive aerosol-rainfall association persisted even when wet scavenging effect was explicitly present in the sampling which strengthen our prime results.

Section 2.5

(3) Underestimation of wet scavenging effect on satellite retrieved AOD values [Grandey et al., 2013;2014].

Section 2.5.3

[revised manuscript text omitted]

11. Section 3.4 could be merged with an earlier description of AOD retrieval errors associated with contamination due to RH. This section is a repetition.

Response: As suggested we have reorganized Section 3.4 as subsection 3.3.2

**Minor Comments**

1. Title: 'Association' instead of 'Associations'.

Response: We have modified this word.

2. Abstract, Line 31: Change to 'Simulated microphysics also illustrated that the…'

Response: We have revised as suggested.

3. Abstract, Line 36: Correct as: 'While the meteorological variability influences'

Response: We have corrected the word.

4. Abstract, Line 37: Change to 'association' instead of 'associations'.

Response: We have corrected it as suggested.

5. Page 2, Line 9: Remove comma (,) after 'cloud base'.

Response: We have removed the comma as suggested.

6. Page 3, Lines 4 & 15; Page 9, Line 20, and many places: Correct 'AP Khain et al.' to 'Khain et al.'.

Response: We have modified the references as suggested.

7. Page 4, Line 2: Replace 'as well as' by 'and'.

Response: We have modified as suggested.

8. Reference required: '…lower available spatial resolution (i.e. 0.25o×0.25o) was in general biased to smaller clouds..'.

Response: We have removed the sentence.

9. Page 9, line 1: Correct: 'CLOUD-aerosol Lidar and infrared pathfinder SATellite (not CloudSat, but CALIPSO)'.

Response: Corrected

10. Page 22, line 18: Correct: 'droplet spectral'.

Response: Corrected at Page 26 Line 11

11. Page 30, Lines 5: Remove 'other'.

Response: Removed

12. Page 33, Lines 5: Change to 'found to be in-line…'

Response: Changed at Page 39 Line 9.

---

## Author Response (AR2)

**Letter to co-editor**

Dear Dr Yun Qian,

Please find below our response to the comments of Reviewer 2.

We have provided detailed clarification regarding the total number of sample points in the figure panels. We have also explained that our methodology of using percentiles to make AOD bins for correlation analysis ensures that equal number of samples are averaged in all the AOD bins from very clean to very polluted scenario. Further, we have also modified the caption of figures 3, 4 and 11 in revised manuscript for better clarity on this issue.

Yours Sincerely,

Prof. S. N. Tripathi

**Author's Response to Reviewer #2**

**Comment:** I thought that the response to the sample number was unclear. They claimed that they had used equal number of samples in each AOD bin, but they still do not want to give the exact number of samples in each AOD bin. The 'n' is not exactly the same, so the number of original sample cannot be the same for each AOD bin. Does the equal number of samples refer to the 50 bins? I guess the number of samples in medium is the biggest, since this is the most common case compared to the very clean and the very polluted cases. Based on the concern of the representation of averaged value for AOD bins in very clean and very polluted cases, I think it is not acceptable for final publication in ACP.

**Response**

Each panel of figure 3 has 50 scatter points, and each scatter point is the average of equal number of data samples (i.e. 'n'; which is 2 percentile of the entire data). 'n' is mentioned in respective panels of figure 3.Thus, the total number of data samples in each panel of figure 3 is equal to 'n' multiplied by 50. For example, for Panel A of figure 3, the total number of collocated data samples (of AOD and RF) used to create 50 scatter points are 50×518 = 25900, in other words, each scatter point in panel A of figure 3 is the average of 518 data samples. Similarly, for Panel B of figure 3, the total number of collocated data samples (of AOD and PR) used to create 50 scatter points are 50 ×111 = 5550 and for Panel C/D/E of figure 3, the total number of collocated data samples (of AOD and CF/CTP/CTP) used to create 50 scatter points are 50 ×674 = 33700.

The reviewer's concerns about disproportional averaging of data samples would have been valid only if the AOD range (0-1) has been divided into 50 linear bins (each of width 0.02) which would have certainly led to different number of data samples in various AOD bins. Instead, our methodology of creating 50 scatter points of equal percentiles (i.e. each AOD bin constitute of 2 percentiles of the total number of data samples) ensures that each AOD bin is the average of equal number of data samples. Many previous studies have also used this methodology for aerosol-cloud associations (for instance *Koren et al.*, 2014 and*Koren et al.*, 2010b). We have further revised the caption of figures 3, 4 and 11 for better clarity.

[revised manuscript text omitted]